# The Geometry of Categorical and Hierarchical Concepts in Large Language Models

**Kiho Park, Yo Joong Choe, Yibo Jiang, and Victor Veitch**
University of Chicago

## Abstract

The linear representation hypothesis is the informal idea that semantic concepts are encoded as linear directions in the representation spaces of large language models (LLMs). Previous work has shown how to make this notion precise for representing binary concepts that have natural contrasts (e.g., {male, female}) as *directions* in representation space. However, many natural concepts do not have natural contrasts (e.g., whether the output is about an animal). In this work, we show how to extend the formalization of the linear representation hypothesis to represent features (e.g., is_animal) as *vectors*. This allows us to immediately formalize the representation of categorical concepts as polytopes in the representation space. Further, we use the formalization to prove a relationship between the hierarchical structure of concepts and the geometry of their representations. We validate these theoretical results on the Gemma and LLaMA-3 large language models, estimating representations for 900+ hierarchically related concepts using data from WordNet.[1]

## 1 Introduction

Understanding how high-level semantic meaning is encoded in the representation spaces of large language models (LLMs) is a fundamental problem in interpretability. A particularly promising avenue is the *linear representation hypothesis* (e.g., Mikolov et al., 2013; Elhage et al., 2022; Nanda et al., 2023; Gurnee & Tegmark, 2024; Park et al., 2024). This is the informal hypothesis that semantic concepts are represented *linearly* in the representation spaces of LLMs. To assess the validity of this hypothesis, and systematically build tools on top of it, we must make precise what it means for a concept to be linearly represented, and understand how semantics are encoded in (the geometry of) the representation spaces.

Focusing on the final softmax layer, Park et al. (2024) give a formalization for the case of binary concepts that can be defined by counterfactual pairs of words. For example, the concept male $\Rightarrow$ female is formalized using the counterfactual pairs {("man", "woman"), ("king", "queen"), ... }. They prove that such binary concepts have a well-defined linear representation as a direction in the representation space. They further connect semantic structure and representation geometry by showing that, under a suitable inner product, *causally separable* concepts that can be freely manipulated (e.g., male $\Rightarrow$ female and french $\Rightarrow$ english) are represented by orthogonal directions.

However, this formalization is limited: many natural concepts cannot be defined by counterfactual pairs of words. For example, simple binary features (e.g., is_animal) or categorical concepts (e.g., {mammal, bird, reptile, fish}) do not admit such formalizations. Additionally, it is not clear how semantic relationships beyond causal separability are encoded in the representation space. In particular, we are interested in this paper in understanding how hierarchical relationships between concepts are encoded in the representation space. That is, what is the relationship between the representations of animal, mammal, and dog?

In this paper, we extend the linear representation hypothesis as follows:

1. We show how to move from representations of binary concepts as *directions* to representations as *vectors*. As a straightforward consequence, this allows us to represent categorical

---

[1]Code is available at github.com/KihoPark/LLM_Categorical_Hierarchical_Representations.

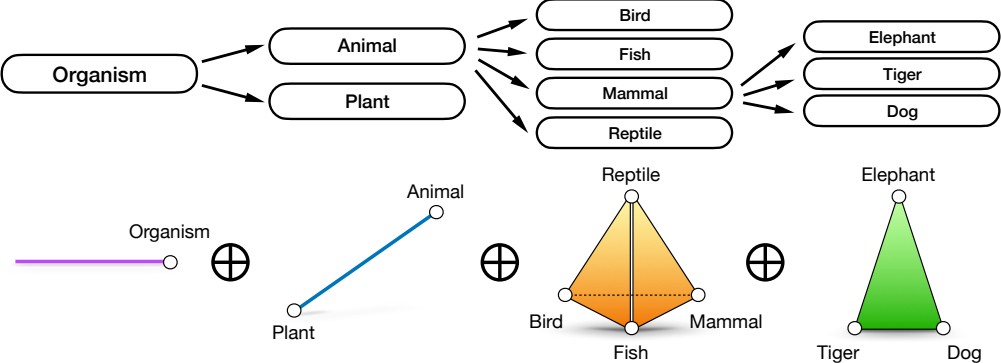

(a) Pictorial depiction of the representation of hierarchically related concepts.

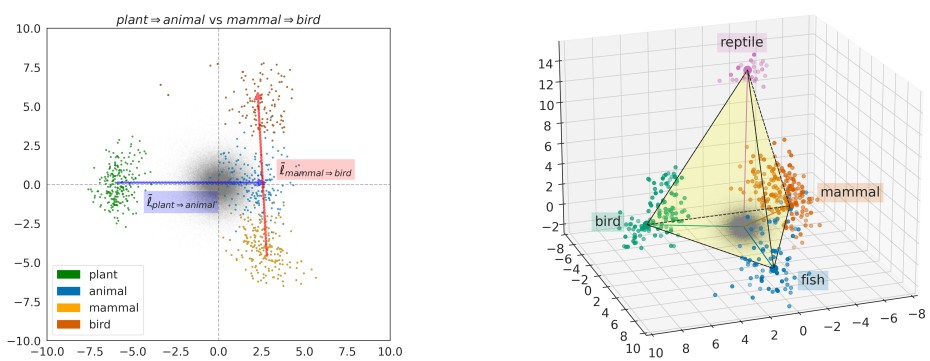

(b) Hierarchy is encoded as orthogonality in Gemma.  (c) Categorical concepts are represented as polytopes in Gemma.

Figure 1: In the representation spaces of LLMs, hierarchically related concepts (such as plant ⇒ animal and mammal ⇒ bird) live in orthogonal subspaces, while categorical concepts are represented as polytopes. The top panel illustrates the structure; the bottom panels show the measured representation structure in the Gemma LLM. See Section 5 and Appendix A for details.

    concepts (e.g., {mammal, bird, reptile, fish}) as polytopes where each vertex is the vector representation of one of the elements of the concept (e.g., is_bird).

2. Using this result, we show that semantic hierarchy between concepts is encoded geometrically as orthogonality between representations, in a manner we make precise.

3. Finally, we empirically validate these theoretical results on the Gemma (Mesnard et al., 2024) and LLaMA-3 (Dubey et al., 2024) LLMs. To that end, we extract concepts from the WordNet hierarchy (Miller, 1995), estimate their representations, and show that the geometric structure of the representations aligns with the semantic hierarchy of WordNet.

The final structure is remarkably simple, and is summarized in Figure 1.

## 2 PRELIMINARIES

We begin with some necessary background.

### 2.1 LARGE LANGUAGE MODELS

For the purposes of this paper, we consider a large language model to consist of two parts. The first part is a function $\lambda$ that maps an input text $x$ to a vector $\lambda(x)$ in a representation space $\Lambda \simeq \mathbb{R}^d$. This is the function given by the stacked transformer blocks. We take $\lambda(x)$ to be the output of the final layer at the final token position. The second part is an unembedding layer that assigns a vector $\gamma(y)$ in an unembedding space $\Gamma \simeq \mathbb{R}^d$ to each token $y$ in the vocabulary. Together, these define a

sampling distribution over tokens via the softmax distribution:

$$\mathbb{P}(y \mid x) = \frac{\exp(\lambda(x)^\top \gamma(y))}{\sum_{y' \in \text{Vocab}} \exp(\lambda(x)^\top \gamma(y'))}. \tag{2.1}$$

We note that the results that follow rely on the duality between the embedding and unembedding spaces, and the softmax link between them. Accordingly, we do not address the "internal" structure of the LLMs. However, we are optimistic that a clear understanding of the softmax geometry will shed light on this as well.

## 2.2 Concepts

We formalize a concept as a latent variable $W$ that is caused by the context $X$ and causes the output $Y$. That is, a concept is a thing that could—in principle—be manipulated to affect the output of the language model. In the particular case where a concept is a binary variable with a word-level counterfactual, we can identify the variable $W$ with the counterfactual pair of outputs $(Y(0), Y(1))$. Concretely, we can identify male $\Rightarrow$ female with $(Y(0), Y(1)) \in_R \{$("man", "woman"), ("king", "queen"), ("he", "her"), ...$\}$. We emphasize that the notion of a concept as a latent variable that affects the output is more general than the counterfactual binary case.

Given a pair of concept variables $W$ and $Z$, we say that $W$ is *causally separable* with $Z$ if the potential outcome $Y(W = w, Z = z)$ is well-defined for all $w, z$. That is, two variables are causally separable if they can be freely manipulated—e.g., we can change the output language and the sex of the subject freely, so these concepts are causally separable.

## 2.3 Causal Inner Product and Linear Representations

We are trying to understand how concepts are represented. At this stage, there are two distinct representation spaces: $\Lambda$ and $\Gamma$. The former is the space of context embeddings, and the latter is the space of token unembeddings. We would like to unify these spaces so that there is just a single notion of representation.

Park et al. (2024) show how to achieve this unification via a "Causal Inner Product". This is a particular choice of inner product that respects the semantics of language in the sense that the linear representations of (binary, counterfactual) causally separable concepts are orthogonal under the inner product. Their result can be understood as saying that there is some invertible matrix $A$ and constant vector $\bar{\gamma}_0$ such that, if we transform the embedding and unembedding spaces as

$$g(y) \leftarrow A(\gamma(y) - \bar{\gamma}_0), \quad \ell(x) \leftarrow A^{-\top}\lambda(x) \tag{2.2}$$

then the Euclidean inner product in the transformed spaces is the causal inner product, and the Riesz isomorphism between the embedding and unembedding spaces is simply the usual vector transpose operation. We can estimate $A$ as the whitening operation for the unembedding matrix. Following this transformation, we can think of the embedding and unembedding spaces as the same space, equipped with the Euclidean inner product.[2]

Notice that the softmax probabilities (eq. (2.1)) are unchanged for any $A$ and $\bar{\gamma}_0$, so this transformation does not affect the model's behavior. The vector $\bar{\gamma}_0$ defines an origin for the unembedding space, and can be chosen arbitrarily. We give a particularly convenient choice below.

In this unified space, the linear representation of a binary concept $W \in_R \{0, 1\}$ is defined as:

**Definition 1.** A vector $\bar{\ell}_W$ is a linear representation of a binary concept $W$ if for all contexts $\ell$, and all concept variables $Z$ that are causally separable with $W$, we have, for all $\alpha > 0$,

$$\mathbb{P}(W = 1 \mid \ell + \alpha\bar{\ell}_W) > \mathbb{P}(W = 1 \mid \ell), \text{ and} \tag{2.3}$$

$$\mathbb{P}(Z \mid \ell + \alpha\bar{\ell}_W) = \mathbb{P}(Z \mid \ell). \tag{2.4}$$

That is, the linear representation is a direction in the representation space that, when added to the context, increases the probability of the concept, but does not affect the probability of any off-target

---

[2]We are glossing over some technical details here; see Park et al. (2024) for details.

concept. The representation is merely a direction because $\alpha \bar{\ell}_W$ is also a linear representation for any $\alpha > 0$ (i.e., there is no notion of magnitude). In the case of concepts corresponding to counterfactual pairs of words, this direction can be shown to be proportional to the "linear probing" direction, and proportional to $g(Y(1)) - g(Y(0))$ for any counterfactual pair $Y(1), Y(0)$ that differ on $W$.

## 3 GENERAL CONCEPTS AND HIERARCHICAL STRUCTURE

Our high-level strategy will be to build up from binary concepts to more complex structure. We begin by defining the basic building blocks.

**Binary and Categorical Concepts**   The most general concept we address in this paper is a categorical concept, which refers to any concept corresponding to a categorical latent variable. This includes binary concepts as a special case. We consider two kinds of binary concept: *binary features* and *binary contrasts*. A binary feature $W \in_R \{\texttt{not\_w}, \texttt{is\_w}\}$ is an indicator of whether the output has the attribute $w$. For example, if the feature $\texttt{is\_animal}$ is true, then the output will be about an animal. A binary contrast $\texttt{a} \Rightarrow \texttt{b} \in_R \{a, b\}$ is a binary variable that contrasts two specific attribute values. For example, the variable $\texttt{mammal} \Rightarrow \texttt{bird}$ is a binary contrast. In the particular case where the binary contrast can correspond to counterfactual pairs of words (e.g., $\texttt{male} \Rightarrow \texttt{female}$), the concept matches the definition used in Park et al. (2024).

**Hierarchical Structure**   The next step is to define what we mean by a hierarchical relation between concepts. To that end, to each attribute $w$, we associate a set of tokens $\mathcal{Y}(w)$ that have the attribute. For example, $\mathcal{Y}(\texttt{mammal}) = \{\text{" dog", " cats", " Tiger"}, \dots\}$. Then,

**Definition 2.** A value $z$ is *subordinate* to a value $w$ (denoted by $z \prec w$) if $\mathcal{Y}(z) \subseteq \mathcal{Y}(w)$. We say a categorical concept $Z \in_R \{z_0, \dots, z_{n-1}\}$ is subordinate to a categorical concept $W \in_R \{w_0, \dots, w_{m-1}\}$ if there exists a value $w_Z$ of $W$ such that each value $z_i$ of $Z$ is subordinate to $w_Z$.

For example, the binary contrast $\texttt{dog} \Rightarrow \texttt{cat}$ is subordinate to the binary feature $\{\texttt{is\_mammal}, \texttt{not\_mammal}\}$, and the binary contrast $\texttt{parrot} \Rightarrow \texttt{eagle}$ is subordinate to the categorical concept $\{\texttt{mammal}, \texttt{bird}, \texttt{fish}\}$. On the other hand, $\texttt{dog} \Rightarrow \texttt{eagle}$ is not subordinate to $\texttt{bird} \Rightarrow \texttt{mammal}$, and $\texttt{bird} \Rightarrow \texttt{mammal}$ and $\texttt{live\_in\_house} \Rightarrow \texttt{live\_in\_water}$ are not subordinate to each other.

**Linear Representations of Binary Concepts**   Now we return to the question of how binary concepts are represented. A key desideratum is that if $\bar{\ell}_W$ is a linear representation, then moving the context embedding in this direction should modify the probability of the target concept *in isolation*. If adding $\bar{\ell}_W$ also modified off-target concepts, it would not be natural to identify it with the target concept $W$. In Definition 1, this idea is formalized by the requirement that the probability of causally separable concepts is unchanged when the representation is added to the context.

We now observe that, when there is hierarchical structure, this requirement is not strong enough to capture 'off-target' behavior. For example, if $\bar{\ell}_{\texttt{animal}}$ captures the concept of animal vs not-animal, then moving in this direction should not affect the relative probability of the output being about a mammal versus a bird. If it did, then the representation would actually capture some amalgamation of the animal and mammal concepts. Accordingly, we must strengthen our definition:

**Definition 3.** A vector $\bar{\ell}_W$ is a linear representation of a binary concept $W$ if

$$\mathbb{P}(W = 1 \mid \ell + \alpha \bar{\ell}_W) > \mathbb{P}(W = 1 \mid \ell), \text{ and} \tag{3.1}$$

$$\mathbb{P}(Z \mid \ell + \alpha \bar{\ell}_W) = \mathbb{P}(Z \mid \ell), \tag{3.2}$$

for all contexts $\ell$, all $\alpha > 0$, and all concept variables $Z$ that are either subordinate to or causally separable with $W$. Here, if $W$ is a binary feature for an attribute $w$, then $W = 1$ denotes $W = \texttt{is\_w}$.

Notice that, in the case of binary contrasts defined by counterfactual pairs, this definition is equivalent to Definition 1, because such variables have no subordinate concepts.

## 4 REPRESENTATIONS OF COMPLEX CONCEPTS

We now turn to how complex concepts are represented. The high-level strategy is to show how to represent binary features as vectors, show how geometry encodes semantic composition, and then use this to construct representations of complex concepts.

### 4.1 VECTOR REPRESENTATIONS OF BINARY AND CATEGORICAL CONCEPTS

To build up to complex concepts, we need to understand how to compose linear representations of binary features. At this stage, the representations are only *directions* in the representation space—they do not have a natural notion of magnitude. In particular, this means we cannot use vector operations (such as addition) to compose representations. To overcome this, we now show how to associate a magnitude to the linear representation of a binary feature. The key is the following result, which connects binary feature representations and word unembeddings:

**Theorem 4** (Magnitudes of Linear Representations). *Suppose there exists a linear representation (normalized direction) $\bar{\ell}_W$ of a binary feature $W$ for an attribute $w$. Then, there is a constant $b_w > 0$ and a choice of unembedding space origin $\bar{\gamma}_0^w$ in eq. (2.2) such that*

$$\begin{cases} \bar{\ell}_W^\top g(y) = b_w & \text{if } y \in \mathcal{Y}(w) \\ \bar{\ell}_W^\top g(y) = 0 & \text{if } y \notin \mathcal{Y}(w). \end{cases} \tag{4.1}$$

*Further, if there exist $d$ attributes $\{w_0, \ldots, w_{d-1}\}$ such that the linear representations of the binary features for these attributes are linearly independent, we can choose a canonical origin $\bar{\gamma}_0$ in eq. (2.2).*

All proofs are given in Appendix B.

The theorem says that, if a (perfect) linear representation of the `animal` feature exists, then every token having the animal attribute has the *same* dot product with the representation vector; i.e., "cat" is exactly as much `animal` as "dog" is. If this weren't true, then increasing the probability that the output is about an animal would also increase the relative probability that the output is about a dog rather than a cat. In practice, such exact representations are unlikely to be found by gradient descent in LLM training. Rather, we expect $\bar{\ell}_W^\top g(y)$ to be isotropically distributed around $b_w$ and 0, with variances that are small compared to $b_w$ (so that animal and non-animal words are well-separated.)

With this result in hand, we can define a notion of vector representation for **binary features**:

**Definition 5.** We say that a binary feature $W$ for an attribute $w$ has a *vector representation $\bar{\ell}_w \in \mathbb{R}^d$* if $\bar{\ell}_w$ satisfies Definition 3 and $\|\bar{\ell}_w\|_2 = b_w$ in Theorem 4. If the vector representation of a binary feature is not unique, we say $\bar{\ell}_w$ is the vector representation that maximizes $b_w$.

We have now moved from representations as directions to representations as vectors. Definition 5 and Theorem 4 give a simple way of composing binary features into **binary contrasts**:

**Corollary 6** (Binary Contrasts Are Vector Differences of Binary Features). *Let $w_0 \Rightarrow w_1$ be a binary contrast, and suppose there exist vector representations $\bar{\ell}_{w_0}$ and $\bar{\ell}_{w_1}$ for each attribute $w_0$ and $w_1$. Then, the difference $\bar{\ell}_{w_1} - \bar{\ell}_{w_0}$ is a linear representation $\bar{\ell}_{w_0 \Rightarrow w_1}$ in the sense of Definition 3.*

We can also apply ordinary vector space operations to construct representation of **categorical concepts**, e.g., $\{$`mammal`, `reptile`, `bird`, `fish`$\}$. There is now a straightforward way to define the representation of such concepts:

**Definition 7.** The *polytope representation* of a categorical concept $W = \{w_0, \ldots, w_{k-1}\}$ is the convex hull of the vector representations of the elements of the concept.[3]

In Appendix D, we state and prove a generalization of Corollary 6 to categorical concepts (which proves Corollary 6 itself).

---

[3]In Appendix C, we prove that the polytope representation for a "natural" categorical concept is a simplex.

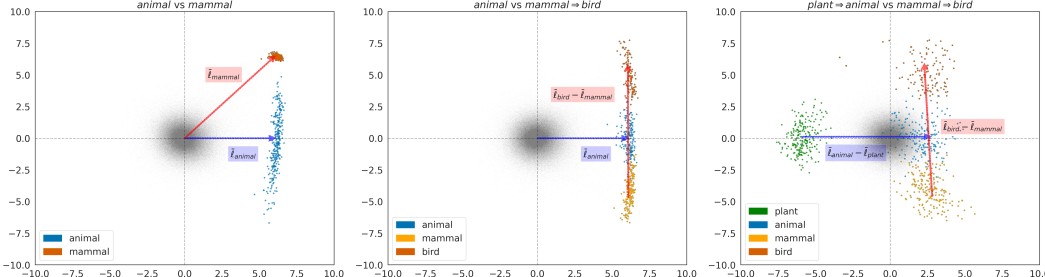

Figure 2: Hierarchical semantics are encoded as orthogonality in the representation space (Theorem 8). The plots show the projection of the unembedding vectors onto 2D subspaces: $\mathrm{span}\{\bar{\ell}_{\texttt{animal}}, \bar{\ell}_{\texttt{mammal}}\}$ (left; (a)), $\mathrm{span}\{\bar{\ell}_{\texttt{animal}}, \bar{\ell}_{\texttt{bird}} - \bar{\ell}_{\texttt{mammal}}\}$ (middle; (b)), and $\mathrm{span}\{\bar{\ell}_{\texttt{animal}} - \bar{\ell}_{\texttt{plant}}, \bar{\ell}_{\texttt{bird}} - \bar{\ell}_{\texttt{mammal}}\}$ (right; (c)). Gray points indicate all 256K tokens in the vocabulary, and the colored points are the tokens in $\mathcal{Y}(w)$. The blue and red vectors are used to span the 2D subspaces.

## 4.2 HIERARCHICAL ORTHOGONALITY

Now, we turn to the question of how hierarchical relationships between concepts are encoded in the representation space. The core intuition is that manipulating the "animal" concept should not affect relative probabilities of the "mammal" and "bird" concepts, so we might expect the representations of `animal` and `mammal` ⇒ `bird` to be orthogonal. The following result formalizes this intuition by connecting the vector and semantic structures. The result is illustrated in Figure 2.

**Theorem 8** (Hierarchical Orthogonality). *Suppose there exist the vector representations for all the following binary features. Then, we have that*

*(a) $\bar{\ell}_w \perp \bar{\ell}_z - \bar{\ell}_w$ for $z \prec w$;*

*(b) $\bar{\ell}_w \perp \bar{\ell}_{z_1} - \bar{\ell}_{z_0}$ for $Z \in_R \{z_0, z_1\}$ subordinate to $W \in_R \{\texttt{not\_w}, \texttt{is\_w}\}$;*

*(c) $\bar{\ell}_{w_1} - \bar{\ell}_{w_0} \perp \bar{\ell}_{z_1} - \bar{\ell}_{z_0}$ for $Z \in_R \{z_0, z_1\}$ subordinate to $W \in_R \{w_0, w_1\}$; and*

*(d) $\bar{\ell}_{w_1} - \bar{\ell}_{w_0} \perp \bar{\ell}_{w_2} - \bar{\ell}_{w_1}$ for $w_2 \prec w_1 \prec w_0$.*

Together, Definition 7 and Theorem 8 give the simple structure illustrated in Figure 1: hierarchical concepts are represented as direct sums of polytopes. This direct sum structure is immediate from Theorem 8. We emphasize that all of the results—involving differences of representations—are only possible because we have *vector* representations (mere directions would not suffice).

## 5 EXPERIMENTS

We now turn to empirically testing the theoretical results. Here, we present our empirical results on the Gemma-2B model (Mesnard et al., 2024).[4] In Appendix F, we further present empirical results on the LLaMA-3-8B model (Dubey et al., 2024), for which our findings are largely analogous.

### 5.1 SETUP

**Canonical Representation** The results in this paper rely on transforming the representation spaces so that the Euclidean inner product is a causal inner product, aligning the embedding and unembedding representations. Following Park et al. (2024), we estimate the required transformation as:

$$g(y) = \mathrm{Cov}(\gamma)^{-1/2}(\gamma(y) - \mathbb{E}[\gamma]) \qquad (5.1)$$

where $\gamma$ is the unembedding vector of a word sampled uniformly from the vocabulary. Centering by $\mathbb{E}[\gamma]$ is a reasonable approximation of centering by $\bar{\gamma}_0$ defined in Theorem 4 because this makes the projection of a random $g(y)$ on an arbitrary direction close to 0. This matches the requirement that the projection of a word onto a concept the word does not belong to should be close to 0.

---

[4]Code is available at github.com/KihoPark/LLM_Categorical_Hierarchical_Representations.

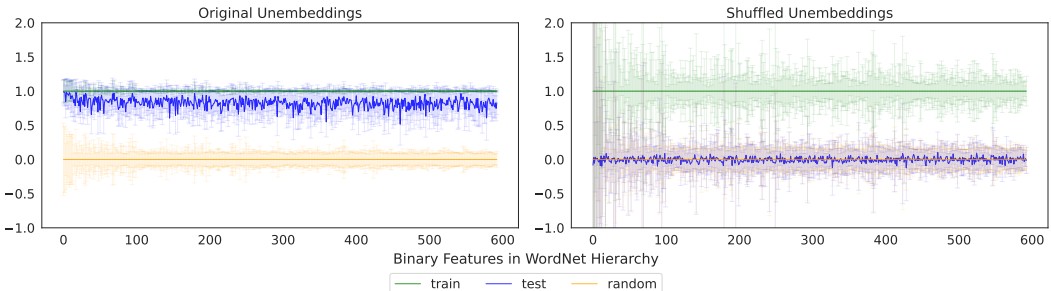

Figure 3: Vector representations exist for most binary features in the WordNet noun hierarchy. For each synset $w$ (indexed on the $x$-axis) we estimate the vector representation $\bar{\ell}_w$ using a train subset of the vocabulary $\mathcal{Y}(w)$. The plot shows the projections $(g(y)^\top \bar{\ell}_w)/\|\bar{\ell}_w\|_2^2$ of train (green), test (blue), and random (orange) words on estimated vector representations for each WordNet feature, using either the original (left) or shuffled (right) unembeddings. Our theory predicts that this value should be close to 1 when $y$ has the target feature, and close to 0 when it does not. The thick lines present the mean of the projections for each feature and the error bars indicate the standard deviation. As predicted, the projections of test words are near 1, and random words near 0 (left plot). Further, this structure does not hold when using the shuffled control without natural semantics (right plot).

**WordNet**   We define a large collection of binary concepts using WordNet (Miller, 1995). WordNet organizes English words into a hierarchy of synsets, where each synset is a set of synonyms. The WordNet hierarchy is based on word hyponym relations, and reflects the semantic hierarchy of interest in this paper. We take each synset as an attribute $w$ and define $\mathcal{Y}(w)$ as the collection of all words belonging to any synset that is a descendant of $w$. For example, the synset `mammal.n.01` is a descendant of `animal.n.01`, so both $\mathcal{Y}(\texttt{mammal.n.01})$ and $\mathcal{Y}(\texttt{animal.n.01})$ contain the word "dog". We collect all noun and verb synsets, and augment the word collections by including plural forms of the nouns, multiple tenses of each verb, and capital and lower case versions of each word. We filter to include only those synsets with at least 50 words in the Gemma vocabulary. This leaves us with 593 noun and 364 verb synsets, each defining an attribute.

For space, we report results on the noun hierarchy here and defer the verb hierarchy to Appendix F.

**Estimation via Linear Discriminant Analysis**   Now, we want to estimate the vector representation $\bar{\ell}_w$ for each attribute $w$. To do this, we make use of vocabulary sets $\mathcal{Y}(w)$. Following Theorem 4, the vector associated to the concept $w$ should have two properties. First, when the full vocabulary is projected onto this vector, the words in $\mathcal{Y}(w)$ should be well-separated from the rest of the vocabulary. Second, the projection of the unembedding vectors for $y \in \mathcal{Y}(w)$ should be approximately the same value. Equivalently, the variance of the projection of the unembedding vectors for $y \in \mathcal{Y}(w)$ should be small. To capture these requirements, we estimate the directions using a variant of Linear Discriminant Analysis (LDA), which finds a projection minimizing within-class variance and maximizing between-class variance. Formally, we estimate the vector representation of a binary feature $W$ for an attribute $w$ as

$$\bar{\ell}_w = \left(\tilde{g}_w^\top \mathbb{E}(g_w)\right) \tilde{g}_w, \quad \text{with} \quad \tilde{g}_w = \frac{\text{Cov}(g_w)^\dagger \mathbb{E}(g_w)}{\|\text{Cov}(g_w)^\dagger \mathbb{E}(g_w)\|_2}, \tag{5.2}$$

where $g_w$ is the unembedding vector of a word sampled uniformly from $\mathcal{Y}(w)$ and $\text{Cov}(g_w)^\dagger$ is a pseudo-inverse of the covariance matrix. We estimate the covariance matrix $\text{Cov}(g_w)$ using the Ledoit-Wolf shrinkage estimator (Ledoit & Wolf, 2004), because the dimension of the representation spaces is much higher than the number of samples.

## 5.2   WORDNET HIERARCHY IS LINEARLY REPRESENTED

**Existence of Vector Representations for Binary Features**   The first question is whether vector representations of binary features exist. To evaluate this, for each synset $w$ in WordNet we split $\mathcal{Y}(w)$ into train words (70%) and test words (30%), fit the LDA estimator to the train words, and examine the projection of the unembedding vectors for the test and random words onto the estimated vector

Table 1: Adding the linear representation of the parent concept to context embeddings does not affect the logit differences between token pairs in a child concept, and it substantially affects those in the parent concept. For each parent-child pair of binary concepts $(w_0 \Rightarrow w_1, z_0 \Rightarrow z_1)$, we first add the normalized linear representation $\bar{\ell}_W = \bar{\ell}_{w_1} - \bar{\ell}_{w_0}$ to the context embeddings. Then, we show the change in logit difference $\log \frac{\mathbb{P}(y_1 \mid \ell)}{\mathbb{P}(y_0 \mid \ell)}$ between the pairs $(y_0, y_1) \in \mathcal{Y}(w_0) \times \mathcal{Y}(w_1)$ (parent; top row) and $(y_0, y_1) \in \mathcal{Y}(z_0) \times \mathcal{Y}(z_1)$ (child; bottom row). Notice that, for any context $x$ and any tokens $y_0$ and $y_1$, adding a linear representation $\bar{\ell}_W$ for a binary contrast $W = w_0 \Rightarrow w_1$ manipulates the logit difference between the tokens from $\ell(x)^\top (g(y_1) - g(y_0))$ to $(\ell(x) + \bar{\ell}_W)^\top (g(y_1) - g(y_0))$, which implies the change in logit difference between the tokens is $\bar{\ell}_W^\top (g(y_1) - g(y_0))$, irrespective of $\ell(x)$. We show the mean and standard deviation of the change in logit differences over all token pairs.

| $W = \text{parent}_0 \Rightarrow \text{parent}_1$ & $Z = \text{child}_0 \Rightarrow \text{child}_1$ | Change in Logit Differences |
|---|---|
| $W = \texttt{plant.n.02} \Rightarrow \texttt{animal.n.01}$ | $5.1265 \pm 1.1731$ |
| $Z = \texttt{mammal.n.01} \Rightarrow \texttt{reptile.n.01}$ | $-0.0600 \pm 1.2190$ |
| $W = \texttt{fluid.n.02} \Rightarrow \texttt{solid.n.01}$ | $9.8296 \pm 1.1099$ |
| $Z = \texttt{crystal.n.01} \Rightarrow \texttt{food.n.02}$ | $0.3770 \pm 1.5410$ |
| $W = \texttt{scientist.n.01} \Rightarrow \texttt{contestant.n.01}$ | $14.4222 \pm 0.9458$ |
| $Z = \texttt{athlete.n.01} \Rightarrow \texttt{player.n.01}$ | $-0.1545 \pm 1.1426$ |

representation. The left plot in Figure 3 shows the mean and standard deviation of the projections, divided by the magnitude of each estimated $\bar{\ell}_w$. Following Theorem 4, if a vector representation exists for an concept, we would expect the values on the test set to be close to 1, and the values for random words to be close to 0. We see that this is indeed the case, giving evidence that vector representations do exist for these features.

As a baseline, the right plot in Figure 3 shows the same analysis but with the unembedding vectors randomly shuffled. In this case, the test projections are close to 0, which indicates that there are no linear representations. Thus, the existence of the linear representations in the original unembeddings relies on the underlying semantic structure.

**Intervention** Next, we validate that adding the estimated linear representations of a binary contrast changes the target concept without changing other off-target concepts, as required by Definition 3. Table 1 shows the mean and standard deviation of the changes in the logit differences between the pairs from parent or child binary contrasts, after adding the normalized linear representations of the parent binary contrast to context embeddings. The results show that the logit differences change significantly for the target concept, while the off-target concept changes very little.

**Relationship Between the Vector Representations of Binary Features** We now turn to examining whether the similarity between the vector representations of binary features reflects their semantic relation. The direct sum structure predicts that concepts that are close in semantic hierarchy should have similar vector representations. For example, $\texttt{mammal}$ and $\texttt{bird}$ are close in the hyponym graph because they share a common parent ($\texttt{animal}$). Our theory predicts that $\bar{\ell}_{\texttt{bird}}$ and $\bar{\ell}_{\texttt{mammal}}$ share a common component $\bar{\ell}_{\texttt{animal}}$ (the representation for "bird" is the representation for "animal" plus an orthogonal component, and similarly for mammal). This implies that the cosine similarity between $\bar{\ell}_{\texttt{mammal}}$ and $\bar{\ell}_{\texttt{bird}}$ should be substantial.

In Figure 4, we show the shortest distance between each feature in the (undirected) WordNet noun hyponym graph in the left panel and the cosine similarity between the estimated vector representations $\bar{\ell}_w$ in the middle panel. It is clear that, as predicted, the cosine similarity reflects the WordNet structure. As a control, we apply the same analysis after randomly shuffling the embeddings and show the resulting cosine similarities in the right panel. Here, direct parent-child (or grandparent-grandchild) relationships are still reflected in the cosine similarity because the shuffled embeddings still respect set inclusion (the words assigned to "mammal" are still a subset of those assigned to "animal"). However, sibling relationships are not reflected once the semantic structure is removed by the shuffling. In this case, the representations of siblings are effectively pairs of random vectors, and are nearly orthogonal as we would expect in a high dimensional space.

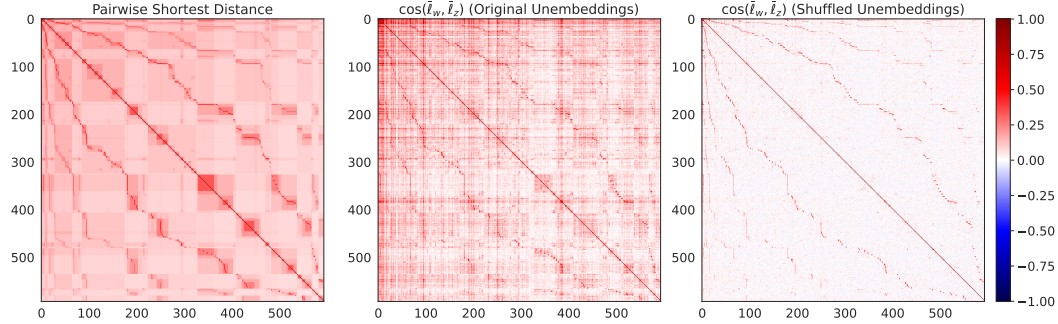

Figure 4: Hierarchical semantics in WordNet are linearly represented in Gemma-2B. The left heatmap shows pairwise shortest distance matrix between features in the noun hierarchy graph as $(1 + \text{min\_distance})^{-1}$ (higher values indicate closeness, such as in child-parent or sibling relationships). The middle heatmap shows the cosine similarity between the vector representations $\bar{\ell}_w$. As predicted, this similarity reflects the WordNet structure. The right heatmap is a control where the embeddings are randomly shuffled (removing semantic structure). In this case, nearly everything is orthogonal, as expected in high-dimensional space (set inclusion relationships remain due to the estimation procedure). In Appendix F, we include zoomed-in versions of these heatmaps.

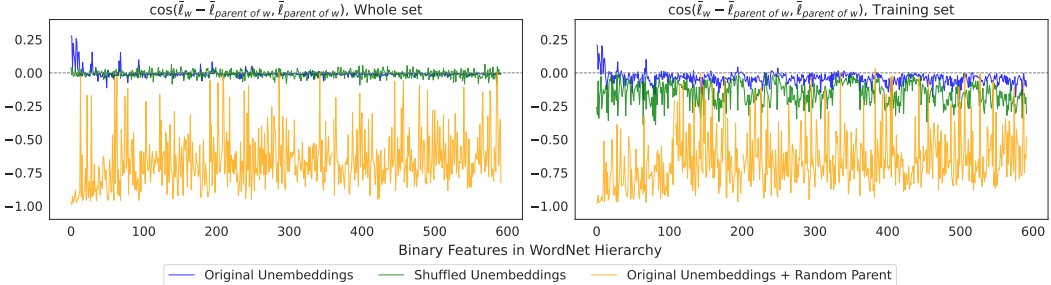

Figure 5: WordNet noun hierarchy is encoded in the orthogonal structure predicted by statement (a) in Theorem 8. We plot the cosine similarity between a child-parent vector and a parent vector for each feature in the hierarchy (blue). As predicted, this value is close to 0. The left plot uses all data for representation estimation, and the right plot uses only 70% independently selected for each synset. We include baselines where a randomly selected feature is used as the parent (orange) and where the embeddings are shuffled (green) as controls for the possibility that the orthogonality is a simple byproduct of high-dimensional geometry, or of the set inclusion relationships used in estimation—see main text for details. See Appendix F for an analogous plot for statement (d).

**Hierarchical Orthogonality**  Finally, we evaluate the prediction that hierarchical relations are encoded orthogonally as predicted in Theorem 8. Figure 5 shows the cosine similarity $\bar{\ell}_{\texttt{parent}}$ and $\bar{\ell}_{\texttt{child}} - \bar{\ell}_{\texttt{parent}}$ for the WordNet features. As predicted, this value is close to 0 (blue curve).

Now, a challenge here is that in high dimensional spaces even random vectors are nearly orthogonal. So, it may be difficult to differentiate whether orthogonality reflects semantic structure or is merely a consequence of nearly everything being orthogonal in high dimensions. As a control, we also show the cosine similarity when the parent vector is the representation of a randomly selected feature (orange). In this case, the cosine similarity is far from 0, suggesting that the observed orthogonality is not merely a byproduct of the high-dimensional geometry.

We also include another baseline that estimates the cosine similarity between the child-parent vector and the parent vector for each feature using shuffled unembeddings (green). In the left plot, we see that we still have orthogonality, which could suggest that the orthogonality is a consequence of the set inclusion relations in WordNet (rather than actual semantic structure).[5] To test this, in the right plot we estimate the representations using only 70% of the tokens, independently selected for each

---

[5]In Appendix G, we explain why set inclusion (before train/test split) leads to orthogonality.

synset. This breaks the set inclusion. In this case, we see the orthogonality is preserved for the original unembeddings, but is broken for the shuffled unembeddings. Note that a cosine of $-0.2$ is highly nontrivial in a high-dimensional space. This suggests the orthogonality does indeed reflect the semantic structure.

## 6  DISCUSSION AND RELATED WORK

We set out to understand how semantic structure is encoded in the geometry of representation space. We have arrived at an astonishingly simple structure, summarized in Figure 1. The key contributions are moving from representing concepts as directions to representing them as vectors (and polytopes), and connecting semantic hierarchy to orthogonality.

**Related Work**   The results here connect closely to the study of linear representations in language models (e.g., Mikolov et al., 2013; Pennington et al., 2014; Arora et al., 2016; Elhage et al., 2022; Burns et al., 2022; Tigges et al., 2023; Nanda et al., 2023; Moschella et al., 2022; Li et al., 2023; Gurnee et al., 2023; Wang et al., 2023; Jiang et al., 2024; Park et al., 2024). In particular, Park et al. (2024) formalize the linear representation hypothesis by unifying three distinct notions of linearity: word2vec-like embedding differences, logistic probing, and steering vectors. Our work relies on this unification, and just focuses on the steering vector notion. Our work also connects to work aimed at theoretically understanding the existence of linear representations. These include early work on word2vec-style embedding models (Arora et al., 2016; Gittens et al., 2017; Arora et al., 2018; Ethayarajh et al., 2018; Frandsen & Ge, 2019; Allen & Hospedales, 2019) as well as dynamic topic models (Blei & Lafferty, 2006; Rudolph et al., 2016). Jiang et al. (2024) connect the existence of linear representations in LLMs to the implicit bias of gradient descent. In this paper, we do not seek to justify the *existence* of linear representations, but rather to understand their *structure* if they do exist. Though, by empirically estimating vector representations for thousands of concepts, we add to the body of evidence supporting the existence of linear representations. Elhage et al. (2022) also empirically observe the formation of polytopes in the representation space of a toy model, and the present work can be viewed in part as giving an explanation for this phenomenon.

There is also a growing literature studying the representation geometry of natural language (Mimno & Thompson, 2017; Reif et al., 2019; Volpi & Malagò, 2021; Li et al., 2020; Chen et al., 2021; Chang et al., 2022; Liang et al., 2022; Jiang et al., 2023; Wang et al., 2023; Park et al., 2024; Valeriani et al., 2024). In terms of hierarchical structures, existing work focuses on connections to hyperbolic geometry (Nickel & Kiela, 2017; Ganea et al., 2018; Chen et al., 2021; He et al., 2024). We do not find such a connection in LLMs, but it is an interesting direction for future work to determine if more efficient LLM representations could be constructed in hyperbolic space. Jiang et al. (2023) hypothesize that very general "independence structures" are naturally represented by partial orthogonality in vector spaces (Amini et al., 2022). The results here confirm and expand on this hypothesis in the case of hierarchical structure in language models.

**Implications and Future Work**   The results in this paper give a foundational understanding the structure of representation space in language models. Of course, the ultimate purpose of foundations is to build upon them. One immediate direction is to refine the attempts to interpret LLM structure to explicitly account for hierarchical semantics. As an example, there is currently significant interest in using sparse autoencoders to extract interpretable features from LLMs (e.g., Cunningham et al., 2023; Bricken et al., 2023; Kissane et al., 2024; Braun et al., 2024). This work searches for representations in terms of distinct binary features. Concretely, it hopes to find features for, e.g., `animal`, `mammal`, `bird`, etc. Based on the results here, these representations are strongly co-linear, and potentially difficult to disentangle. On the other hand, a representation in terms of $\bar{\ell}_{\texttt{animal}}$, $\bar{\ell}_{\texttt{mammal}} - \bar{\ell}_{\texttt{animal}}$, $\bar{\ell}_{\texttt{bird}} - \bar{\ell}_{\texttt{animal}}$, etc., will be cleanly separated and equally interpretable. Fundamentally, semantic meaning has hierarchical structure, so interpretability methods should respect this structure. Understanding the geometric representation makes it possible to design such methods.

In a separate, foundational, direction: the results in this paper rely on using the canonical representation space, and we estimate this using the whitening transformation of the unembedding layer. However, this technique only works for the final layer representation. It is an important open question how to make sense of the geometry of internal layers.

ACKNOWLEDGMENTS

This work is supported by ONR grant N00014-23-1-2591 and Open Philanthropy.

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

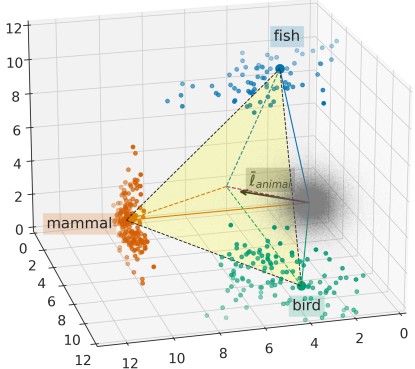 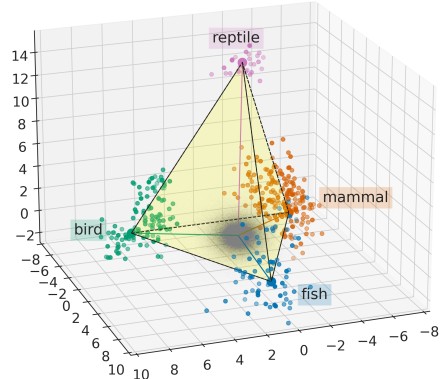

Figure 6: Categorical concepts are represented as polytopes. The plots show the projection of the unembedding vectors on the 3D subspaces: $\text{span}\{\bar{\ell}_{\texttt{mammal}}, \bar{\ell}_{\texttt{bird}}, \bar{\ell}_{\texttt{fish}}\}$ (left) and $\text{span}\{\bar{\ell}_{\texttt{bird}} - \bar{\ell}_{\texttt{mammal}}, \bar{\ell}_{\texttt{fish}} - \bar{\ell}_{\texttt{mammal}}, \bar{\ell}_{\texttt{reptile}} - \bar{\ell}_{\texttt{mammal}}\}$ (right). The gray points indicate all 256K tokens in the vocabulary, and the colored points are the tokens in $\mathcal{Y}(\texttt{w})$. The left plot further shows the orthogonality between the triangle and the projection of $\bar{\ell}_{\texttt{animal}}$ (black arrow).

## A  VISUALIZATION OF `animal`

As a concrete example for visualizations, we examine the theoretical predictions for the familiar concept `animal`. However, the categories in WordNet are not the most intuitive; for example, the subcategories under texttt{animal.n.01} include terms like "chordate", "aquatic vertebrate", "invertebrate" instead of more familiar categories such as "fish", "amphibian", or "insect". This makes it challenging to gather tokens corresponding to clear, high-level concepts $\{\texttt{mammal}, \texttt{bird}, \texttt{fish}, \texttt{reptile}, \texttt{amphibian}, \texttt{insect}\}$. Therefore, we generated two sets of tokens $\mathcal{Y}(\texttt{animal})$ and $\mathcal{Y}(\texttt{plant})$ using ChatGPT-4 (OpenAI, 2023), and manually inspected them. $\mathcal{Y}(\texttt{animal})$ is further divided into six sets of tokens for each subcategory $\{\texttt{mammal}, \texttt{bird}, \texttt{fish}, \texttt{reptile}, \texttt{amphibian}, \texttt{insect}\}$.

Figure 2 illustrates the geometric relationships between various representation vectors. The main takeaway is that the semantic hierarchy is encoded as orthogonality in the manner predicted by Theorem 8. The figure also illustrates Theorem 4, showing that the projection of the unembedding vectors for $y \in \mathcal{Y}(w)$ is approximately constant, while the projection of $y \notin \mathcal{Y}(w)$ is zero.

Figure 6 illustrates that the representation of a categorical concept is a polytope. and the projection of unembedding vectors onto the subspace for the polytope are concentrated on each vertex as predicted in Corollary 10. The left plot also shows that, as predicted, the polytope for $\{\texttt{fish}, \texttt{mammal}, \texttt{bird}\}$ is orthogonal to the vector representation of `animal`.

## B  PROOFS

### B.1  PROOF OF THEOREM 4

**Theorem 4** (Magnitudes of Linear Representations). *Suppose there exists a linear representation (normalized direction) $\bar{\ell}_W$ of a binary feature $W$ for an attribute $w$. Then, there is a constant $b_w > 0$ and a choice of unembedding space origin $\bar{\gamma}_0^w$ in eq. (2.2) such that*

$$\begin{cases} \bar{\ell}_W^\top g(y) = b_w & \text{if } y \in \mathcal{Y}(w) \\ \bar{\ell}_W^\top g(y) = 0 & \text{if } y \notin \mathcal{Y}(w). \end{cases} \tag{4.1}$$

*Further, if there exist $d$ attributes $\{w_0, \ldots, w_{d-1}\}$ such that the linear representations of the binary features for these attributes are linearly independent, we can choose a canonical origin $\bar{\gamma}_0$ in eq. (2.2).*

*Proof.* For any $y_1, y_0 \in \mathcal{Y}(w)$ or $y_1, y_0 \notin \mathcal{Y}(w)$, let $Z$ be a binary concept where $\mathcal{Y}(Z = 0) = \{y_0\}$ and $\mathcal{Y}(Z = 1) = \{y_1\}$. Since $Z$ is subordinate to $W$, eq. (3.2) implies that

$$\text{logit } \mathbb{P}(Y = y_1 \mid Y \in \{y_0, y_1\}, \ell + \bar{\ell}_W) = \text{logit } \mathbb{P}(Y = y_1 \mid Y \in \{y_0, y_1\}, \ell) \quad \text{(B.1)}$$

$$\iff \bar{\ell}_W^\top(g(y_1) - g(y_0)) = \bar{\ell}_W^\top A(\gamma(y_1) - \gamma(y_0)) = 0 \quad \text{(B.2)}$$

where $A$ is the invertible matrix in eq. (2.2). This means that $\bar{\ell}_W^\top A\gamma(y)$ is the same for all $y \in \mathcal{Y}(w)$, and it is also the same for all $y \notin \mathcal{Y}(w)$.

Furthermore, for any $y_1 \in \mathcal{Y}(w)$ and $y_0 \notin \mathcal{Y}(w)$, eq. (3.1) implies that

$$\text{logit } \mathbb{P}(Y = y_1 \mid Y \in \{y_0, y_1\}, \ell + \bar{\ell}_W) > \text{logit } \mathbb{P}(Y = y_1 \mid Y \in \{y_0, y_1\}, \ell) \quad \text{(B.3)}$$

$$\iff \bar{\ell}_W^\top(g(y_1) - g(y_0)) = \bar{\ell}_W^\top A(\gamma(y_1) - \gamma(y_0)) > 0. \quad \text{(B.4)}$$

Thus, by setting $b_w^0 = \bar{\ell}_W^\top A\gamma(y)$ for any $y \notin \mathcal{Y}(w)$, and $b_w = \bar{\ell}_W^\top A\gamma(y_1) - \bar{\ell}_W^\top A\gamma(y_0) > 0$ for any $y_1 \in \mathcal{Y}(w)$ and $y_0 \notin \mathcal{Y}(w)$, we get

$$\begin{cases} \bar{\ell}_W^\top A\gamma(y) = b_w^0 + b_w & \text{if } y \in \mathcal{Y}(w) \\ \bar{\ell}_W^\top A\gamma(y) = b_w^0 & \text{if } y \notin \mathcal{Y}(w). \end{cases} \quad \text{(B.5)}$$

Then, we can choose an origin as

$$\bar{\gamma}_0^w = b_w^0 A^{-1} \bar{\ell}_W \quad \text{(B.6)}$$

satisfying eq. (4.1).

On the other hand, if there exist $d$ attributes $\{w_0, \ldots, w_{d-1}\}$ such that the linear representations $\bar{\ell}_{W_0}, \ldots, \bar{\ell}_{W_{d-1}}$ of the binary features $W_0, \ldots, W_{d-1}$ for these attributes are linearly independent, then the linear system

$$\bar{\ell}_{W_i}^\top A\bar{\gamma}_0 = b_{w_i}^0 \quad \text{for} \quad i = 0, \ldots, d - 1 \quad \text{(B.7)}$$

has a unique solution $\bar{\gamma}_0$. We can choose this vector as the canonical origin $\bar{\gamma}_0$ in eq. (2.2), ensuring that eq. (4.1) is satisfied. $\qquad\square$

### B.2 PROOF OF THEOREM 8

**Theorem 8** (Hierarchical Orthogonality). *Suppose there exist the vector representations for all the following binary features. Then, we have that*

(a) $\bar{\ell}_w \perp \bar{\ell}_z - \bar{\ell}_w$ *for* $z \prec w$;

(b) $\bar{\ell}_w \perp \bar{\ell}_{z_1} - \bar{\ell}_{z_0}$ *for* $Z \in_R \{z_0, z_1\}$ *subordinate to* $W \in_R \{\texttt{not\_w}, \texttt{is\_w}\}$;

(c) $\bar{\ell}_{w_1} - \bar{\ell}_{w_0} \perp \bar{\ell}_{z_1} - \bar{\ell}_{z_0}$ *for* $Z \in_R \{z_0, z_1\}$ *subordinate to* $W \in_R \{w_0, w_1\}$; *and*

(d) $\bar{\ell}_{w_1} - \bar{\ell}_{w_0} \perp \bar{\ell}_{w_2} - \bar{\ell}_{w_1}$ *for* $w_2 \prec w_1 \prec w_0$.

*Proof.* (a) For $\bar{\ell}_w$ and $\bar{\ell}_z$ where $z \prec w$, by Theorem 4, we have

$$\begin{cases} (\bar{\ell}_z - \bar{\ell}_w)^\top g(y) = b_z - b_w & \text{if } y \in \mathcal{Y}(z) \\ (\bar{\ell}_z - \bar{\ell}_w)^\top g(y) = 0 - b_w = -b_w & \text{if } y \in \mathcal{Y}(w) \setminus \mathcal{Y}(z) \\ (\bar{\ell}_z - \bar{\ell}_w)^\top g(y) = 0 - 0 = 0 & \text{if } y \notin \mathcal{Y}(w). \end{cases} \quad \text{(B.8)}$$

When $w \setminus z$ denotes an attribute defined by $\mathcal{Y}(w) \setminus \mathcal{Y}(z)$, $\bar{\ell}_z - \bar{\ell}_w$ can change the target concept $w \setminus z \Rightarrow z$ without changing any other concept subordinate or causally separable to the target concept. Thus, $\bar{\ell}_z - \bar{\ell}_w$ is the linear representation $\bar{\ell}_{w \setminus z \Rightarrow z}$. This concept means $\texttt{not\_z} \Rightarrow \texttt{is\_z}$ conditioned on $w$, and hence it is subordinate to $w$.

Therefore, $\bar{\ell}_w$ is orthogonal to the linear representation $\bar{\ell}_{w \setminus z \Rightarrow z} = \bar{\ell}_z - \bar{\ell}_w$ by the property of the causal inner product. If they are not orthogonal, adding $\bar{\ell}_w$ can change the other concept $w \setminus z \Rightarrow z$, and it is a contradiction.

(b) By the above result (a), $\bar{\ell}_w^\top(\bar{\ell}_{z_1} - \bar{\ell}_w) = \bar{\ell}_w^\top(\bar{\ell}_{z_0} - \bar{\ell}_w) = 0$. Therefore, $\bar{\ell}_w^\top(\bar{\ell}_{z_1} - \bar{\ell}_{z_0}) = 0$.

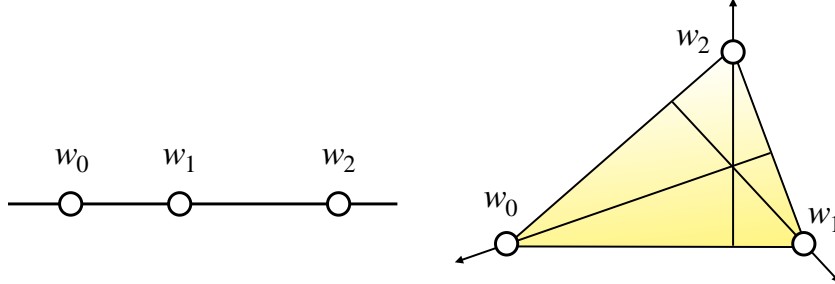

Figure 7: Illustration of the case $k = 3$ in the proof of Proposition 9.

(c) Let's say that $w_1$ is $w_Z$ defined in Definition 2. The binary contrast $z_0 \Rightarrow z_1$ is subordinate to the binary feature for the attribute $w_0$. By the property of the causal inner product, $\bar{\ell}_{w_0}$ is orthogonal to the linear representation $\bar{\ell}_{z_0 \Rightarrow z_1} = \bar{\ell}_{z_1} - \bar{\ell}_{z_0}$ (by Corollary 10). Then, with the above result (b), we have $(\bar{\ell}_{w_1} - \bar{\ell}_{w_0})^\top (\bar{\ell}_{z_1} - \bar{\ell}_{z_0})$.

(d) By the above result (a), we have

$$\begin{cases} \|\bar{\ell}_{w_1} - \bar{\ell}_{w_0}\|_2^2 = \|\bar{\ell}_{w_1}\|_2^2 - \|\bar{\ell}_{w_0}\|_2^2 \\ \|\bar{\ell}_{w_2} - \bar{\ell}_{w_1}\|_2^2 = \|\bar{\ell}_{w_2}\|_2^2 - \|\bar{\ell}_{w_1}\|_2^2 \\ \|\bar{\ell}_{w_2} - \bar{\ell}_{w_0}\|_2^2 = \|\bar{\ell}_{w_2}\|_2^2 - \|\bar{\ell}_{w_0}\|_2^2. \end{cases} \tag{B.9}$$

Then,

$$\|\bar{\ell}_{w_1} - \bar{\ell}_{w_0}\|_2^2 + \|\bar{\ell}_{w_2} - \bar{\ell}_{w_1}\|_2^2 \tag{B.10}$$

$$= \|\bar{\ell}_{w_1}\|_2^2 - \|\bar{\ell}_{w_0}\|_2^2 + \|\bar{\ell}_{w_2}\|_2^2 - \|\bar{\ell}_{w_1}\|_2^2 \tag{B.11}$$

$$= \|\bar{\ell}_{w_2}\|_2^2 - \|\bar{\ell}_{w_0}\|_2^2 \tag{B.12}$$

$$= \|\bar{\ell}_{w_2} - \bar{\ell}_{w_0}\|_2^2. \tag{B.13}$$

Therefore, $\bar{\ell}_{w_1} - \bar{\ell}_{w_0}$ is orthogonal to $\bar{\ell}_{w_2} - \bar{\ell}_{w_1}$.

$\square$

## C  NATURAL CATEGORICAL CONCEPTS AS SIMPLICES

Polytopes are quite general objects. Definition 7 also includes representations of categorical variables that are semantically unnatural, e.g., $\{\texttt{dog}, \texttt{sandwich}, \texttt{running}\}$. We would like to make a more precise statement about the representation of "natural" concepts. One possible notion of a "natural" concept is one where the model can freely manipulate the output values. The next proposition shows such concepts have particularly simple structure:

**Proposition 9** (Categorical Concepts are Represented as Simplices). *Suppose that $\{w_0, \ldots, w_{k-1}\}$ is a collection of $k$ mutually exclusive attributes such that for every joint distribution $Q(w_0, \ldots w_{k-1})$ there is some $\ell_i$ such that $\mathbb{P}(W = w_i \mid \ell_i) = Q(W = w_i)$ for every $i$. Then, the vector representations $\bar{\ell}_{w_0}, \ldots, \bar{\ell}_{w_{k-1}}$ form a $(k-1)$-simplex in the representation space. In this case, we take the simplex to be the representation of the categorical concept $W = \{w_0, \ldots, w_{k-1}\}$.*

*Proof.* If we can represent arbitrary joint distributions, this means, in particular, that we can change the probability of one attribute without changing the relative probability between a pair of other attributes. Consider the case where $k = 3$, as illustrated in Figure 7. If $\bar{\ell}_{w_0}, \bar{\ell}_{w_1}, \bar{\ell}_{w_2}$ are on a line, then there is no direction in that line (to change the value in the categorical concept) such that adding the direction can change the probability of $w_2$ without changing the relative probabilities between $w_0$ and $w_1$. However, if $\bar{\ell}_{w_0}, \bar{\ell}_{w_1}, \bar{\ell}_{w_2}$ are not on a line, they form a triangle. Then, there exists a line

that is toward $\bar{\ell}_{w_2}$ and perpendicular to the opposite side of the triangle. Now adding the direction $\tilde{\ell}$ can manipulate the probability of $w_2$ without changing the relative probabilities between $w_0$ and $w_1$. That is, for any $\alpha > 0$ and context embedding $\ell$,

$$
\begin{cases}
\mathbb{P}(W = w_2 \mid \ell + \alpha\tilde{\ell}) > \mathbb{P}(W = w_2 \mid \ell), \text{ and} \\
\frac{\mathbb{P}(W=w_1 \mid \ell+\alpha\tilde{\ell})}{\mathbb{P}(W=w_0 \mid \ell+\alpha\tilde{\ell})} = \frac{\mathbb{P}(W=w_1 \mid \ell)}{\mathbb{P}(W=w_0 \mid \ell)}.
\end{cases}
\tag{C.1}
$$

Therefore, the vectors $\bar{\ell}_{w_0}, \bar{\ell}_{w_1}, \bar{\ell}_{w_2}$ form a 2-simplex.

This argument extends immediately to higher $k$ by induction. For each $i \in \{0, \ldots, k-1\}$, there should exist a direction that is toward $\bar{\ell}_{w_i}$ and orthogonal to the opposite hyperplane $((k-2)$-simplex) formed by the other $\bar{\ell}_{w_{i'}}$'s. Then, the vectors $\bar{\ell}_{w_0}, \ldots, \bar{\ell}_{w_{k-1}}$ form a $(k-1)$-simplex. $\qquad\square$

## D  SUBSPACES FOR CATEGORICAL CONCEPTS

By Theorem 4, polytope representations have the following property:

**Corollary 10** (Polytope Representations). *Let $W = \{w_0, \ldots, w_{k-1}\}$ be a categorical concept, and suppose there exist vector representations $\bar{\ell}_{w_i}$ for the binary features of each attribute $w_i$ (their convex hull is the polytope representation of $W$). Let $\bar{L} = \left[\bar{\ell}_{w_1} - \bar{\ell}_{w_0}, \ldots, \bar{\ell}_{w_{k-1}} - \bar{\ell}_{w_0}\right] \in \mathbb{R}^{d \times (k-1)}$ and let $\mathcal{C}(\bar{L})$ be its column space. Then, there exist some vectors $\tau_{w_i} \in \mathcal{C}(\bar{L})$, for each $i$, such that*

$$
\begin{cases}
\prod_{\bar{L}} g(y) = \tau_{w_i} & \text{if } y \in \mathcal{Y}(w_i) \\
\prod_{\bar{L}} g(y) = 0 & \text{if } y \notin \cup_{i=0}^{k-1} \mathcal{Y}(w_i),
\end{cases}
\tag{D.1}
$$

*where $\prod_{\bar{L}} = \bar{L}(\bar{L}^\top \bar{L})^\dagger \bar{L}^\top$ is the projection matrix onto $\mathcal{C}(\bar{L})$, and $(\bar{L}^\top \bar{L})^\dagger$ denotes a pseudo-inverse of the matrix.*

*Proof.* By Theorem 4, for $i = 1, \ldots, k-1$, we have

$$
\prod_{\bar{L}} g(y) = \bar{L}(\bar{L}^\top \bar{L})^\dagger \bar{L}^\top g(y) = \bar{L}(\bar{L}^\top \bar{L})^\dagger \left[0, \ldots, 0, b_{w_i}^2, 0, \ldots, 0\right]^\top := \tau_{w_i}
\tag{D.2}
$$

for any $y \in \mathcal{Y}(w_i)$. Similarly, we have

$$
\prod_{\bar{L}} g(y) = \bar{L}(\bar{L}^\top \bar{L})^\dagger \bar{L}^\top g(y) = \bar{L}(\bar{L}^\top \bar{L})^\dagger \left[-b_{w_0}^2, \ldots, -b_{w_0}^2\right]^\top := \tau_{w_0}
\tag{D.3}
$$

for any $y \in \mathcal{Y}(w_0)$, while

$$
\prod_{\bar{L}} g(y) = \bar{L}(\bar{L}^\top \bar{L})^\dagger \bar{L}^\top g(y) = \bar{L}(\bar{L}^\top \bar{L})^\dagger [0, \ldots, 0]^\top = 0
\tag{D.4}
$$

for any $y \notin \cup_{i=0}^{k-1} \mathcal{Y}(w_i)$. Therefore, we have eq. (D.1) with some constant vectors $\tau_{w_i}$. $\qquad\square$

In words, the unembeddings of tokens for the same attribute share the same projections on the subspace spanned by the differences between the vector representations. Therefore, adding any vector in the subspace $\mathcal{C}(\bar{L})$ to the context embedding changes the probability of the target concept $W = \{w_0, \ldots, w_{k-1}\}$ *without* changing any other concept subordinate to or causally separable with the target concept. Finally, note that Corollary 10 implies Corollary 6.

## E  EXPERIMENT DETAILS

We employ the `Gemma-2B` version of the Gemma model (Mesnard et al., 2024), which is accessible online via the `huggingface` library. Its two billion parameters are pre-trained on three trillion tokens. This model utilizes 256K tokens and 2,048 dimensions for the representation space.

We always use tokens that start with a space ('\u2581') in front of the word, as they are used for next-word generation with full meaning. Additionally, like WordNet data we use, we include plural

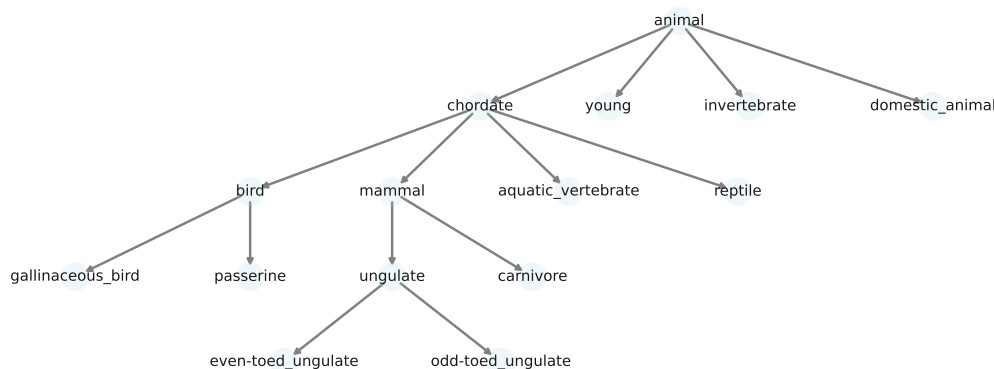

Figure 8: Subtree in WordNet noun hierachy for descendants of `animal`.

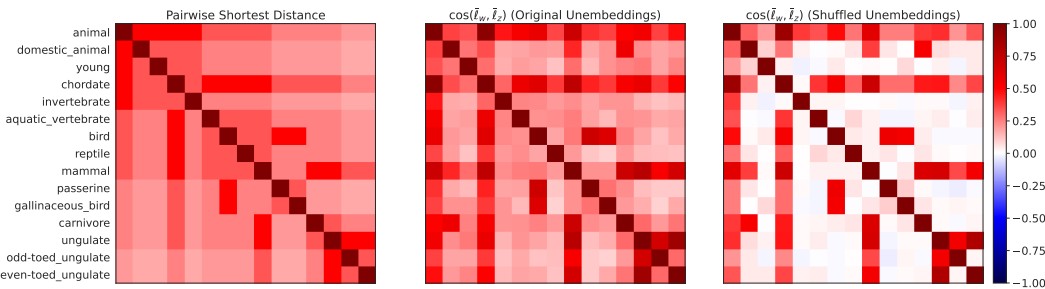

Figure 9: Zoomed-in Heatmaps of the subtree for `animal` in Figure 8.

forms, and both capital and lowercase versions of the words in $\mathcal{Y}(\text{animal})$ and $\mathcal{Y}(\text{plant})$ for visualization in Appendix A.

In the WordNet synset data, each content of the synset `mammal.n.01` indicates that "mammal" is a word, "n" denotes "noun," and "01" signifies the first meaning of the word. In the WordNet hierarchy, if a parent has only one child, we combine the two features into one. Additionally, since the WordNet hierarchy is not a perfect tree, a child can have more than one parent. We use one of the parents when computing the $\bar{\ell}_w - \bar{\ell}_{\text{parent of } w}$.

Lastly, we use the vector representations estimated by the non-split collection of tokens for Figure 2, Table 1, Figure 4, Figure 6, Figure 14, and Figure 18.

# F    ADDITIONAL RESULTS

**Zooming in on a Subtree of Noun Hierarchy**    As it is difficult to understand the entire WordNet hierarchy at once from the heatmaps in Figure 4, we present a zoomed-in heatmap for the subtree (Figure 8) for the feature `animal` in Figure 9. The left heatmap displays the shortest distance between the nodes of the subtree in Figure 8. The middle heatmap shows that the cosine similarities between the vector representations $\bar{\ell}_w$ reflect the child-parent or sibling relationships. The right heatmap demonstrates that the vector representations estimated by shuffled unembeddings only reflect the set inclusion relationships, and other pairs have cosine similarities close to 0.

**Additional Results on Hierarchical Orthogonality**    Analogous to Figure 5, we validate the statement (d) in Theorem 8. Figure 10 shows that a child-parent vector and a parent-grandparent vector for each feature estimated by the original unembedidngs (blue) are orthogonal. The random parent and grandparent vectors (orange) make the cosine similarity not close to 0, suggesting that the orthogonality is not merely a byproduct of the high-dimensional space.

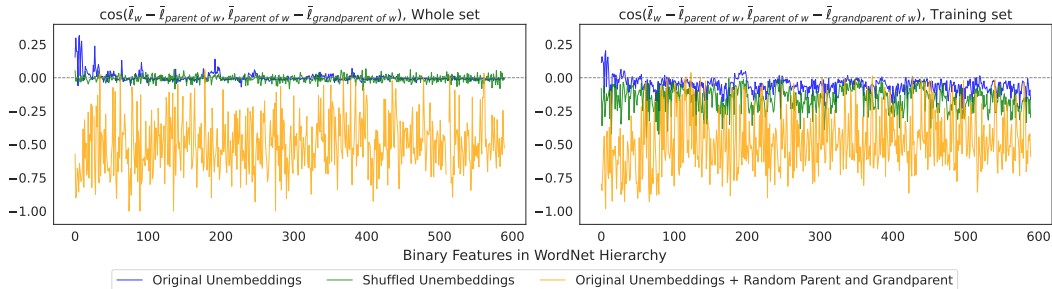

Figure 10: WordNet noun hierarchy is encoded as the orthogonal structure predicted by statement (d) in Theorem 8. The cosine similarity between a child-parent vector and a parent-grandparent vector for each feature in the hierarchy estimated by original (blue) and shuffled (green) unembedding vectors with the whole (left) and training (right) set of tokens. Another baseline (orange) is the cosine similarity with random parent and grandparent vectors.

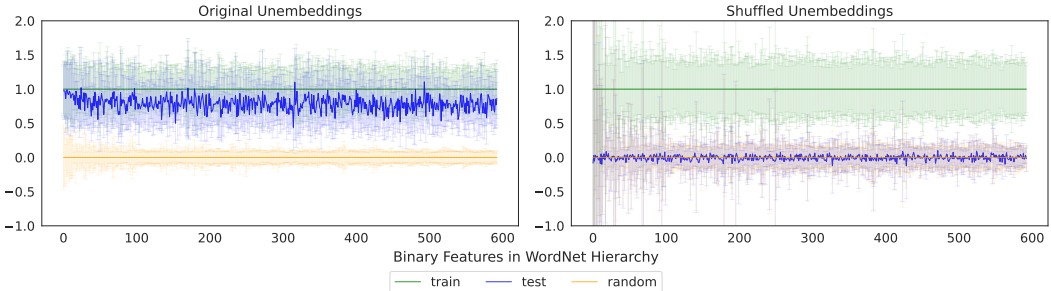

Figure 11: Comparison of projection of train (green), test (blue), and random (orange) words on estimated **mean vector** for each WordNet feature, from the original unembeddings (left) and the shuffled unembeddings (right). The value for each word $y$ is $(g(y)^\top \bar{\ell}_w)/\|\bar{\ell}_w\|_2^2$ that we expect to be 1 when $y$ has the target feature. The $x$-axis indices denote all features in the noun hierarchy. The thick lines present the mean of the projections for each feature and the error bars indicate the standard deviation. The features are ordered by the hierarchy.

Similar to the findings in Figure 5, the shuffled unembedding vectors (green) reflect the set inclusion relationships, and the cosine similarities are close to 0. When we violate the set inclusion by using only the 70% training set of tokens, the right plot in Figure 10 shows that the cosine similarities from shuffled (green) unembeddings are much smaller, while those from original (blue) unembeddings are still close to 0. This indicates that the hierarchical orthogonality is not a trivial consequence of the set inclusion, but rather reflects the semantic hierarchy.

**The Mean Estimator for the Vector Representation Has High Variance**   The mean vector $\mathbb{E}(g_w)$ can serve as an estimator for the vector representation $\bar{\ell}_w$. However, Figure 11, which is analogous to Figure 3, shows that the projections of train and test words onto the mean vector have larger variances. Therefore, the LDA direction (eq. (5.2)) would be a more appropriate estimator for the vector representation.

**Geometry of Hierarchical relations on the Euclidean Inner Product**   In this paper, we transform the representation spaces to use the causal inner product. However, what if we employ the naive Euclidean inner product instead? To address this, we estimate the vector representations for the WordNet noun hierarchy estimated from the original Gemma unembedding vectors through centering alone, without applying whitening (which is necessary for the causal inner product). This approach still preserves the Euclidean inner product. Then, Figure 12 shows that the hierarchical orthogonality is not satisfied in the Euclidean inner product space, as evidenced by the cosine similarities not being close to 0.

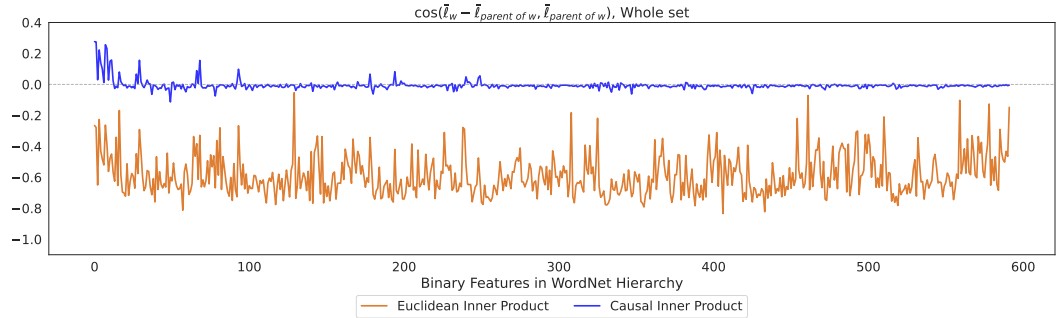

Figure 12: WordNet noun hierarchy is not encoded as the orthogonal structure when we use the naive Euclidean inner product, whereas it is encoded as orthogonality in the causal inner product.

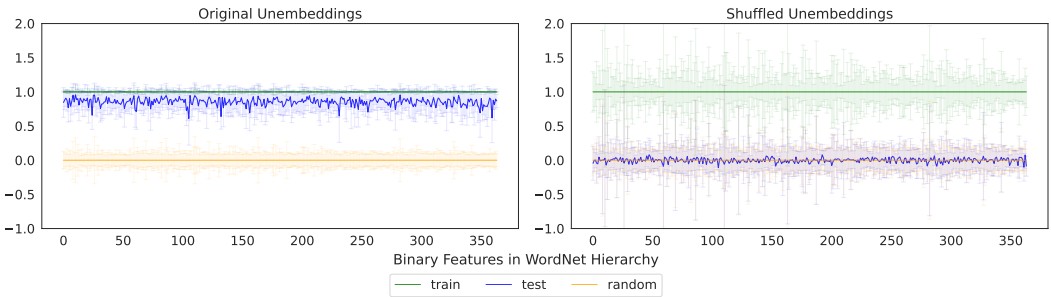

Figure 13: Vector representations exist for most binary features in the WordNet **verb** hierarchy.

**WordNet Verb Hierarchy**   In the same way as for the noun hierarchy, we estimate the vector representations for the WordNet verb hierarchy. Analogous to Figure 3, Figure 13 shows that the vector representations exist for most binary features in the WordNet verb hierarchy. Analogous to Figure 4, Figure 14 shows that the hierarchical semantics in WordNet are encoded in the Gemma representation space. Analogous to Figure 5 and Figure 10, Figure 15 and Figure 16 show that the hierarchical orthogonality is encoded in the Gemma representation space for the WordNet verb hierarchy.

**WordNet Noun Hierarchy encoded in LLaMA-3 Model**   We also validate the theoretical predictions in the LLaMA-3-8B model (Dubey et al., 2024) in the same way as for the Gemma model. The vector representations for most binary features in the WordNet noun hierarchy exist in the LLaMA-3 model (Figure 17). The hierarchical semantics in WordNet are encoded in the LLaMA-3 representation space (Figure 18). Lastly, the hierarchical orthogonality is encoded in the LLaMA-3 representation space for the WordNet noun hierarchy (Figure 19 and Figure 20).

## G   WHY DOES SET INCLUSION GIVE THE ORTHOGONALITY?

The left panel of Figure 5 shows that the cosine similarity between the child-parent vector and the parent vector is also close to 0 when they are estimated by the shuffled unembedding vectors. One of the possible explanations for this phenomenon is that the set inclusion relationships (and the estimating process) give the orthogonality. For example, the set inclusion between the collections $\mathcal{Y}(z) = \{\texttt{dog}, \texttt{sandwich}, \texttt{running}\}$ for a child $z$ and $\mathcal{Y}(w) = \{\texttt{dog}, \texttt{sandwich}, \texttt{running}, \texttt{France}, \texttt{scientist}, \texttt{diamond}\}$ for a parent $w$ can make the child-parent vector and the parent vector orthogonal.

Since the LDA-based vector representation estimated by eq. (5.2) is similar to the mean vector $\mathbb{E}(g_w)$, we briefly derive the orthogonality using the mean vector as the vector representation. Suppose that $a_1, \ldots, a_{N_a}, b_1, \ldots, b_{N_b} \overset{i.i.d.}{\sim} \mathcal{N}(0, I_d)$, which may correspond to the (centered and whitened) shuffled unembeddings. Here, $a_i$'s are for the child $z$, and both $a_i$'s and $b_i$'s are for the par-

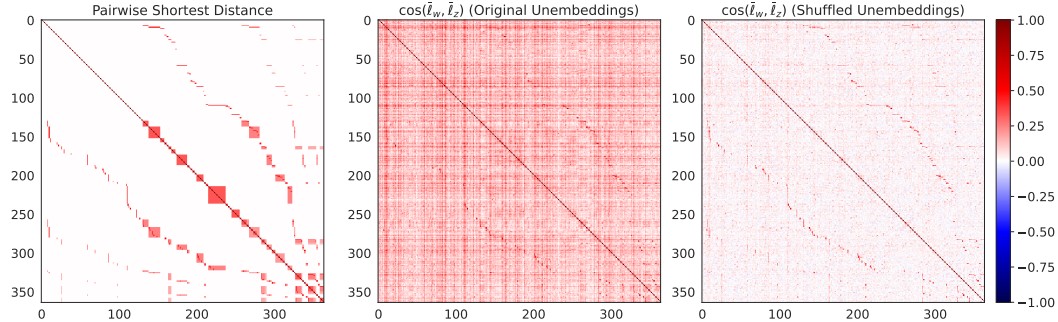

Figure 14: WordNet **verb** hierarchy is encoded in Gemma representation space.

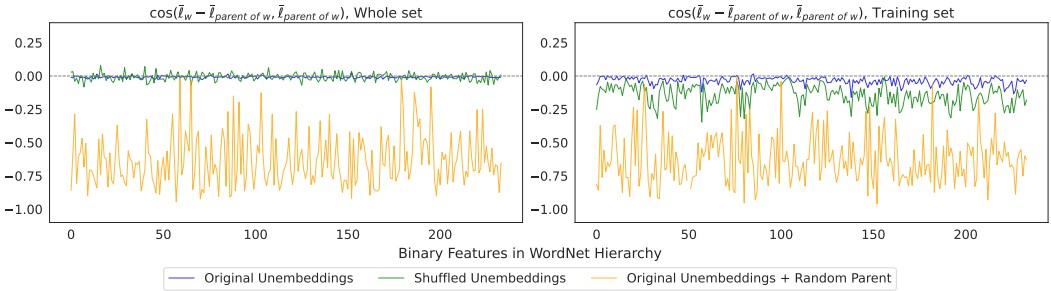

Figure 15: WordNet **verb** hierarchy is encoded as the orthogonal structure predicted by statement (a) in Theorem 8.

ent $w$. Then, the parent vector is $v_w = \frac{1}{N_a + N_b}(\sum a_i + \sum b_i)$, and the child vector is $v_z = \frac{1}{N_a}\sum a_i$. We know that $a_1, \ldots, a_{N_a}, b_1, \ldots, b_{N_b}$ are mutually orthogonal and $\|a_i\|^2 = \|b_i\|^2 = d$ with high probability. Therefore, we have

$$(v_z - v_w)^\top v_w = \tag{G.1}$$

$$= \left(\frac{N_b}{N_a(N_a + N_b)}\sum a_i - \frac{1}{N_a + N_b}\sum b_i\right)^\top \left(\frac{1}{N_a + N_b}\sum a_i + \frac{1}{N_a + N_b}\sum b_i\right) \tag{G.2}$$

$$= \frac{N_b}{N_a(N_a + N_b)^2}(\sum a_i)^\top \sum a_i - \frac{1}{(N_a + N_b)^2}(\sum b_i)^\top \sum b_i \tag{G.3}$$

$$= \frac{N_b}{N_a(N_a + N_b)^2}(N_a d) - \frac{1}{(N_a + N_b)^2}(N_b d) \tag{G.4}$$

$$= 0 \tag{G.5}$$

with high probability.

In words, the orthogonality between the child-parent vector and the parent vector can be derived from the set inclusion relationships. We can say that the set inclusion is the hierarchical relation as defined in Definition 2, but natural semantic hierarchy is our main focus in this paper. In this context, the right panel of Figure 5 shows that the violation of the set inclusion relationships makes the cosine similarity not close to 0, suggesting that semantic hierarchy is encoded as orthogonality.

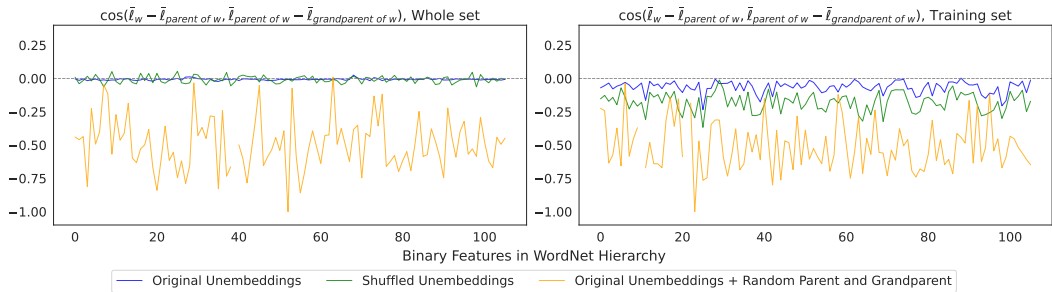

Figure 16: WordNet **verb** hierarchy is encoded as the orthogonal structure predicted by statement (d) in Theorem 8.

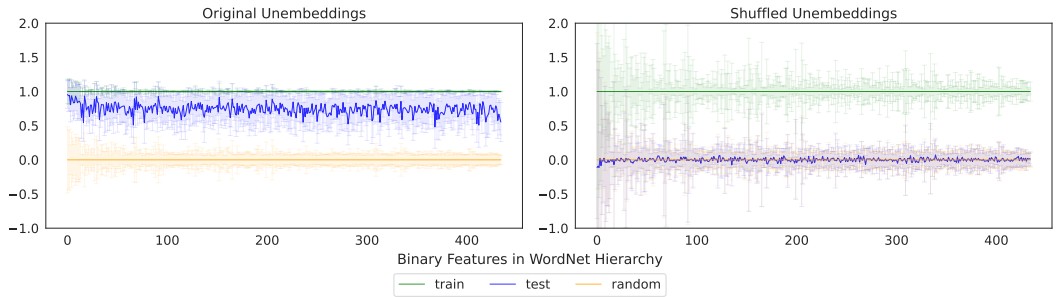

Figure 17: Vector representations for most binary features in the WordNet noun hierarchy exist in the **LLaMA-3** model.

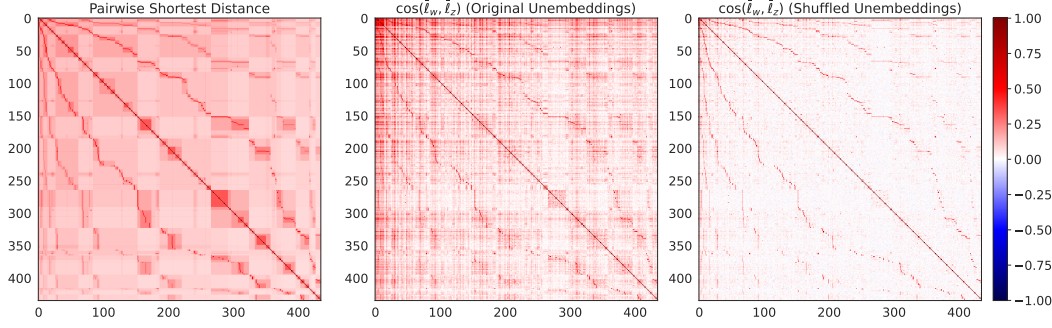

Figure 18: WordNet noun hierarchy is encoded in **LLaMA-3** representation space.

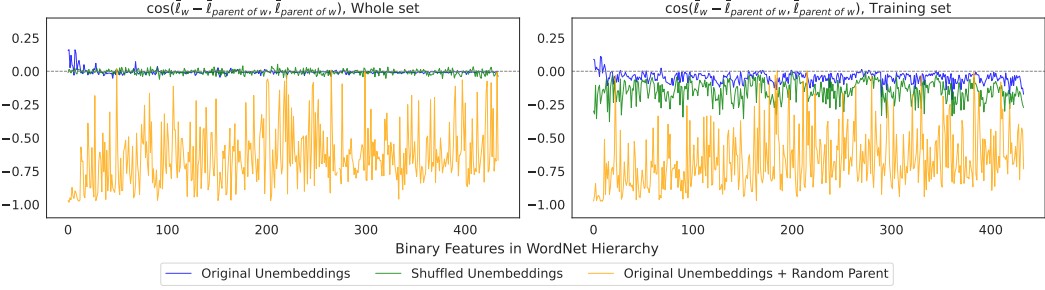

Figure 19: In **LLaMA-3** representation space, WordNet noun hierarchy is encoded as the orthogonal structure predicted by statement (a) in Theorem 8.

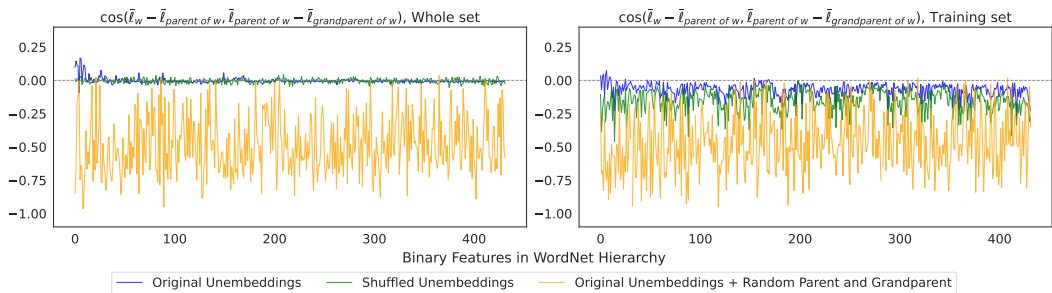

Figure 20: In **LLaMA-3** representation space, WordNet noun hierarchy is encoded as the orthogonal structure predicted by statement (d) in Theorem 8.

