# OpenReview forum: "The Geometry of Categorical and Hierarchical Concepts in Large Language Models"
_ICLR.cc/2025/Conference — ICLR 2025 Oral_

### Official Review · Reviewer_zPtr · 2024-10-16

**Soundness:** 2
**Presentation:** 3
**Contribution:** 2
**Rating:** 5
**Confidence:** 3

**Summary:**

As I understand it, the core value proposition of this paper is the report of an empirical observation: that, with respect to a particular inner-product, conceptual representations in transformer-based LLMs are vectorial, and moreover that hierarchical relationships between concepts correspond to orthogonality, while co-hyponymic categorical concepts are organised as simplices, altogether resulting in a direct-sum of polytopes.

The particular (sort/kind of) inner-product used is called "causal" (CIPs), which established in (apparently immediately) prior work by Park et al. CIPs are a piece of conceptual technology, which I understand as follows:
> Assuming an operational/counterfactual notion of concept as a latent variable in a causal chain <Input -> Latent -> Output>...
> Defining causal separability of two concept-variables to be well-definedness of the output on their independent joint distributions...
> An inner product is "causal" when it treats causally separated concept-variables as orthogonal.
> Park et al. gives methods to obtain and compute with such CIPs, and shows how CIPs admit a duality theorem that permits embedding and unembedding spaces to be identified and viewed as isomorphic to some Euclidean space equipped with the usual inner product (in contexts where there are embedding, softmax, and unembedding maps to speak of.)

**Strengths:**

To me, there are two noteworthy aspects of the empirical claim.

<Emergence> First is that this structure arises emergently and naturally in LLM representations. This would be a scientific contribution to ongoing discourse concerning how and whether LLMs understand [cf. Bender & Koller's "Climbing towards NLU" which casts LLMs as Searle-type non-understanders vs. Sogaard's "Grounding the Vector Space of an Octopus" which surveys evidence that language models obtain geometric models of concepts in their vectorial representations, as the current paper does.]

<Word-Token-Concept> Second is that the claim holds for transformer-based LLMs, where (to my understanding) tokenisation decouples the relationship between words and semantic vectors. Demonstrating empirically that semantic representations not only exist but satisfy geometric constraints according to their natural-language conceptual organisation *at the token level* is big-if-true, even just methodologically.

**Weaknesses:**

This is a simple and compelling observation, especially in the broader context of approaches to vectorial semantics downstream of Firth's distributional hypothesis, where there are many rhyming ideas in the literature. For example, the usage of an inner-product in a high-dimensional space to test for membership in a vectorial data structure is prefigured by Kanerva's hyperdimensional computing. Hyponymic structure alongside convex simplicial structure is treated mathematically in [Interacting Conceptual Spaces I: Grammatical Composition of Concepts; https://arxiv.org/pdf/1703.08314], and the idea of looking for hypo/hypernymic structural representations in vector spaces is its own cottage industry [as a google search of the terms "hyponymy" and "density matrix" will demonstrate.]

While the authors have taken pains to point out (lines 51, 124, 225, 308) that the current approach is a conceptual advance *due to the consideration of vectors* rather than, e.g. directional cones in prior work, this could be read as a strawman that misrepresents a broader research context of vectorial semantics, and it just doesn't do justice to the empirical claims of the paper. Look, even if I totally discount any claim to conceptual novelty regarding the use of vectorial semantics, I still would consider the paper to be strong on the basis of the empirical observation alone, provided the observation is adequately situated.

Integrating some comparison to approaches in vectorial semantics could be an opportunity to strengthen the conceptual dimension of the paper. As far as I can tell, other vectorial semantics worok typically has a "feature-engineering" rather than exploratory nature, and using the geometry of causal inner products seems to be an effective and relatively novel idea; having these as literature supported claims could potentially broaden the appeal and reach of this work at little cost.

**Questions:**

I am currently borderline because I have questions, but I am happy to raise my score if my major questions are addressed: First is what CIPs have to do with LLMs and with the particular empirical observation. Second is how the <Word-Token-Concept> gap was crossed. I'm going to write a little adversarially now, but in hopefully a productive way; I like the paper and would like to strengthen it.

**************

1. Regarding CIPs, their robustness, and their conceptual relevance for the empirical claim at hand.

The content of CIPs was difficult to glean from the materials presented in this paper alone, and I had to refer to the prior work for details. Given the closeness of presentation (down to the choice of variables), I would recommend explicit attribution of credit to the prior work. As a presentational note, just reproducing Thm 3.2 would have gone some way to answering my questions as a reader about why "duality" is used as a term, and why the embedding and unembedding spaces are assumed to be Euclidean and of matching dimension, etc. It is perhaps clearer to indicate that having a CIP is desirable because it enables dual presentation of embeddings and unembeddings, it renders computation of a semantic character as easy as regular Euclidean inner products, and so on.

Here are several cases of conceptual difficulty I had that made it difficult for me to understand what the role of CIP in this story is, and I hope you can help sort me out.

> The definition of CIP from causal separation is potentially too stringent; consider the counterexample case where two co-hyponymic concepts are only defined in the domain of a common hypernym, e.g. "is_bird" and "is_nocturnal" are causally separable properties but only hold in the domain of "is_animal", whereas the current statement requires well-definition of birdness and nocturnality everywhere. So ok, let's say that as a reader I don't trust your first-principles approximation of what a concept is; is it worth spending the time and space to convince me or is that not really important for what needs to be shown here?

> In the mathematical setup of CIP, it is assumed that there is a single euclidean embedding space for text, a matching unembedding space, and a softmax. This is for just the "final token position" (line 107), so I'm assuming an autoregressive transformer, but then there's no further talk of the "internal structure of LLMs" (line 113). This is suspicious to me, because if there's absolutely nothing structurally relevant about the architecture of LLMs to the subsequent analysis of concepts and causal separation and then CIPs, then I can freely swap "LLMs" for anything else? If you're doing something so general-purpose, then I would like to be convinced that CIPs are actually necessary, rather than a contrived thing that the empirical data doesn't really depend upon.

> Where exactly are CIPs used or useful? My best guess so far is that CIPs and their assumptions are required in order to prove Theorem 8, which yields the analytic presentation in Figure 1 as a direct sum of polytopes. However, the polytope-simplices never have their convexity property explored or exploited [cf. Gardenfors "Geometry of Meaning"], and the particular example presented in Figure 1 is a special case where at most one node at every level has children, so I believe a fair equivalent characterisation would also be: "different levels are orthogonal, and co-hyponymic nodes form clusters." I'm not convinced that this is really worth all the baggage of CIP, so where else is it being employed?

2. The Word-Token-Concept gap.

WordNet and vectorial embeddings of words are perfectly good and commonplace. What I was excited to learn about was how you managed to extend this analysis over to the tokens of LLMs. I must have missed something, because the only thing I can see that gives some kind of methological hint as to how this gap was crossed is a sentence in line 955, tucked in the appendix: "Lastly, we use the vector representations estimated by the non-split collection of tokens for Figure 2, (etc.)" I feel like I must have missed it somehow because the omission of this crucial methodology would be such an inconsistency with the attention to detail for the rest of the paper, but I just can't figure out what is being done here to cross-validate findings on WordNet with Gemma and Llama. The mathematical assumptions made only raised more questions (assume final token only for an autoregressive... predictor? autoencoder?).

**************

These two questions are crucial for me because they would make the difference between an uncharitable reading (e.g. CIP is a just-so construct and the empirical work is mostly WordNet with unclear methodology for LLMs), and a very happy one (e.g. CIP is a general-purpose information geometry, and here is a clear and well-motivated demonstration of how this conceptual lens can be applied to find conceptual structure even at the token level of LLMs).

---

> ### Author Response · Authors · 2024-11-25
> **Response Part 1**
>
> Thank you for your amazingly thorough and thoughtful review!
>
> We do not view the main contribution of the paper as empirical. Rather, our goal here is **scientific** in the following sense. The aim is to consider a hypothesis---concepts are `linearly represented’ internally to LLMs---formalize this hypothesis, use the formalization to make falsifiable predictions, and then conduct empirical evaluations of the predictions to see if they are indeed falsified. In this view, the flow of the paper is:
> 1. Begin with the high-level linear representation hypothesis.
> 2. (Definition 3) Formalize this in terms of modification to the output distribution induced by adding a vector in a given direction, and observe that a sensible formalization requires the representation of a concept to *not modify* off-target concepts. This is the first key insight.
> 3. Combine this with the structure of the softmax distribution to derive two consequences of the hypothesis. First, theorem 4, which says that the inner product between any word exhibiting an attribute and the linear representation of that attribute must be constant. Note that this is a theoretical prediction, and is a consequence of the softmax structure! Second, theorem 8, which relates semantic hierarchy to orthogonality in the representation space.
> 4. Then, we see if the hypothesis can be falsified by empirically testing the predictions. We observe strong agreement between the theoretical consequences of the linear representation hypothesis and the actual structure of LLMs. We take this to be evidence for the hypothesis, and its derived consequences.
>
> In particular, our claims are relatively narrow ones about the structure of linear representations *in consequence of the softmax function*. This doesn’t say anything about how concepts are or ought to be represented in systems other than LLMs. Nevertheless, the connections to the broader literature that you point out are fascinating, and we will certainly add pointers to the related work!
>
> Similarly, the emphasis on vector (rather than direction) structure should be understood within the scope of the study of linear representations in large language models specifically. Here, it has been unclear how to assign magnitude, and a main technical contribution of this work is showing that there is indeed a natural way to do so.

---

> > ### Author Response · Authors · 2024-11-25
> > **Response Part 2**
> >
> > With respect to your questions:
> >
> > For the causal inner product, we will further clarify the motivation and choice of CIP. In brief: geometric notions (e.g., norms and orthogonality, which play leading roles in the development) are most naturally defined with respect to an inner product. Accordingly, we need a 'suitable' one for studying the semantic structure of representations. For our purposes, the main important fact is that previous work has already established a particularly natural choice for such an inner product. We very much like your suggestion `"It is perhaps clearer to indicate that having a CIP is desirable because it enables dual presentation of embeddings and unembeddings, it renders computation of a semantic character as easy as regular Euclidean inner products, and so on."` and will use this to motivate the use of the CIP! (Indeed, this is the content of lines 144-148, but your framing is much easier to parse than ours, which invoked the Reisz isomorphism).
> >
> > 1. `"The definition of CIP from causal separation is potentially too stringent…"` This is a great point! The short answer is that it’s not so important to convince the reader that the conceptual underpinnings are ironclad, the development "just" requires you to accept the use of potential outcome notation. The longer answer is that the ambiguity you’re picking up on here is a thing that haunts all causal reasoning. This is described as “the inherent vagueness of all causal questions” in Hernan’s 2016 article “Does water kill?”  The philosophical problem of really precisely specifying what is an intervenable variable is basically unsolved. The trick of the potential outcomes notation is to sidestep this ambiguity enough to allow mathematical reasoning without being hamstrung by definitional issues. We’re basically encountering the same ambiguity that happens in epidemiology and econometrics, and sidestepping it the same way. (In addition to the Hernan article, there’s good writing on this from Frederick Eberhardt.)
> >
> >
> > 2. The key special LLM structure here is the use of the softmax distribution to construct the next word probabilities---equation 2.1. In particular, this means that sentences are represented as finite-dimensional vectors, and word pieces are represented as dual vectors. Note also that we do not explain *why* linear representations arise, rather we say that if linear representations exist then they must have certain properties that we derive. It is unclear whether the “why” is more LLM specific than just the softmax (though we’d guess that the softmax is really the key ingredient).
> >
> > The theoretical development uses the property of CIPs that concepts that can be freely varied will have orthogonal representation. In principle, the Euclidean inner product could have this property (i.e., could be a CIP) and it wouldn’t be necessary to explicitly do a unifying transformation. The evidence that the transformation is necessary is simply that Park et al find naive Euclidean doesn’t respect semantics, and we also found that naive Euclidean doesn’t reflect the geometric structure we derive (see figure 12).
> >
> >
> > 3. The CIP property is necessary to connect 'freely varying' and orthogonality, and is used in theorem 8.
> >
> >
> > Finally, regarding the “word-token-concept” gap: We did not fully understand what you’re asking here---is it possible for you to clarify?
> >
> > Our best guess is that the question is basically about the gap between LLMs as objects that represent *sentences* and WordNet as giving structure over words. If this is the question, then the resolution is simply that, in essence, we are studying the geometry induced by the softmax function, which connects sentence embeddings and word piece unembeddings. We use the wordnet word structure to pull out hierarchy from the words in the LLM vocabulary (see, in particular, the beginning of section 3)
> >
> > (The splitting stuff is mainly a control for potential finite sample issues in the experiments rather than a conceptually key point. The idea is just to evaluate with a separate test set to make sure that the observed structure can’t actually be a consequence of overfitting-like phenomena).

---

> ### Comment · Reviewer_zPtr · 2024-11-27
> **word-token-concept**
>
> I'm happy for now with the other points. Thank you for this request for clarification regarding the "word-token-concept" gap, let me articulate my concern more precisely. My confusion relates to how your findings bridge three distinct levels of representation:
>
> 1. WordNet synsets (word/concept level)
> 2. LLM tokens (subword/character level)
> 3. Semantic concepts (meaning level)
>
> By my understanding, while WordNet gives us structure over words and their meanings, LLMs fundamentally operate on tokens which can be subwords, characters, or other units. You've shown geometric structure in the token space, but I'm looking to understand:
>
> 1. Your methodology for mapping between WordNet concepts and LLM tokens
> 2. How you handle cases where concepts map to multiple tokens
> 3. How you validate that the geometric structures reflect genuine semantic relationships rather than artifacts of token selection
>
> You partially address this by mentioning space-prefixed tokens and including plural/case variations, but I believe the paper would be strengthened by explicitly describing your complete mapping methodology between WordNet concepts and token sets, your approach to handling tokenization effects, and evidence that your geometric findings reflect semantic structure rather than tokenization artifacts.
>
> Would you be able to clarify these aspects? Mindful of the incoming deadline for paper updates, I would be satisfied with an explanation within this discussion format.

---

> > ### Author Response · Authors · 2024-11-27
> > **Response to the reviewer's response**
> >
> > Thank you for clarifying!
> >
> > First, with respect to associating concepts to LLM tokens:
> >
> > The theory associates to each concept a set of words that exhibit that concept; see line  184. For the experiments, we construct such sets using a recursive scheme where for each concept we define the associated set of words to be the union of
> > 1. the words belonging to the child synsets in wordnet (so, the animal set gets {'mammal', 'bird', 'reptile', etc}, but not the word 'animal')
> > 2. the set of words associated to each of the child synsets (so, e.g., if the constructed word set for mammal includes {'poodle', 'leopard', 'marmot', 'dog', etc} then we also include all of these words in the set associated to animal)
> >
> > Then, to translate these sets of words to sets of unembedding vectors we:
> > 1. include all obvious variants of each word (e.g., capitalization, pluralization for nouns, tense for verbs)
> > 2. include each unembedding vector in the set if the word is in the vocabulary (so, e.g., animals that are tokenized into multiple word pieces are excluded)
> >
> > In particular, each concept maps to many tokens (e.g., the concept 'animal' contains 958 tokens). The code for creating this data is included in the supplementary material as `get_wordnet_hypernym_gemma.ipynb`. We will also clarify further the construction in the experiments section.
> >
> > We hope this also clarifies your second question. Empirically, we consider only concepts that are included as named synsets in WordNet (note, however, that these can be quite abstract—e.g., 'social_group.n.01' and ‘mathematical_relation.n.01')
> >
> > Next, with respect to validating that the geometric relationships are not artifacts of token selection:
> >
> > This is the main aim of the controls in the experiments. In brief:
> >
> > 1. We find that the geometry respects semantic structure among sibling relationships that share no common tokens. For example, the sets of words associated to mammal and bird are nearly disjoint (the word ' griffin' is the exception). However, because all birds and all mammals are animals, we should have that the representation of bird and mammal overlap (on the common animal component). Indeed, we do observe such structure—see Figure 4.  Note that this cannot be an artifact of token selection (because of the disjoint structure) and indeed we see the shuffled baseline does not exhibit such sibling similarity.
> > 2. We try randomly shuffling the vocabulary to break semantic structure, and find that the predicted results then do not hold (so, the results we see cannot be attributed simply to the nested set construction)
> > 3. We use test-train splits to separately estimate the representation vectors and measure their predicted properties—see Figures 3 and 5.

---

> > > ### Author Response · Authors · 2024-12-01
> > >
> > > Thank you again for your review!
> > >
> > > We hope that we have addressed your major questions. If so, we hope that you will consider raising your score.

---

### Official Review · Reviewer_yBgV · 2024-10-26

**Soundness:** 3
**Presentation:** 3
**Contribution:** 3
**Rating:** 8
**Confidence:** 3

**Summary:**

This paper investigates the representation of semantic concepts in large language models (LLMs), specifically testing and extending the linear representation hypothesis. This hypothesis proposes that semantic features, like gender or categorical distinctions, can be represented linearly in the embedding space. However, existing representations primarily address binary concepts with natural contrasts. This paper broadens this to include hierarchical and categorical concepts, proposing that these can be represented as vectors and polytopes within the representation space. Additionally, the paper formalizes hierarchical relationships as orthogonality between representations and validates these claims on the Gemma and LLaMA-3 models using WordNet-derived hierarchies.

**Strengths:**

1. Innovative Extension of the Linear Representation Hypothesis: The authors rigorously extend the hypothesis by introducing vector and polytope representations for non-binary, hierarchical concepts. This is particularly valuable because it addresses concepts lacking binary contrasts and presents a way to capture categorical hierarchies in language model embeddings.

2. Mathematical Rigor and Clear Formalization: The introduction of the causal inner product is mathematically grounded, ensuring that causally separable concepts are orthogonal in the transformed space (Equation 2.2). This rigor adds depth to the standard linear representation hypothesis by linking embedding and unembedding spaces without affecting model probabilities (Equation 2.1).
Theorem 4 (on magnitudes of binary representations) and Theorem 8 (on hierarchical orthogonality) are presented with precise definitions and assumptions. The mathematical derivation ensures that vector magnitudes and orthogonal properties align with the semantics of hierarchical structures.

3. Clear and Well-Defined Experimental Setup: The experiments on the WordNet hierarchy using Linear Discriminant Analysis (LDA) are robust. By estimating vector representations for over 900 concepts, the paper empirically tests whether hierarchical concepts are represented orthogonally in the embedding space. Results showing the preservation of hierarchical orthogonality even after shuffling embeddings further validate their claims.

4. Practical Implications: Extending concept representation to polytopes and using orthogonality for hierarchy offers a promising direction for LLM interpretability. The authors’ approach could pave the way for more structured and interpretable embeddings, potentially enhancing downstream tasks such as knowledge extraction and structured generation.

**Weaknesses:**

1. Clarity on the Choice of Transformation in Causal Inner Product: While the causal inner product transformation (Equation 2.2) is mathematically sound, further clarification on how the invertible matrix and constant vector are determined would strengthen the paper’s rigor. Specifically, elaborating on the practical implications of choosing these parameters and their influence on orthogonal representations in various models could provide more insight for researchers replicating or extending this approach.

2. More Explicit Connection to Hierarchical Concepts: The hierarchical orthogonality introduced in Theorem 8 is foundational to the paper’s approach, yet its practical consequences could be explored more thoroughly. For instance, how does this orthogonality affect the interpretability or utility of representations in hierarchical tasks? Including a discussion of possible applications, such as taxonomy creation or hierarchical clustering, would enrich the paper’s contribution and situate it within the broader field of LLM interpretability.

3. Expanding on Limitations and Approximation Issues: The paper notes that exact linear representations may not always emerge during LLM training due to stochasticity in optimization. While this limitation is understandable, the authors could discuss how close approximations might affect the reliability of hierarchical representations, especially under more complex, multi-level hierarchies. This would give readers a clearer view of the limitations of the proposed framework.

**Questions:**

1. On the Selection of Parameters for the Causal Inner Product: Can you elaborate on how you select the matrix A and the vector gamma in the causal inner product, and whether this choice generalizes across models? Would different values of A significantly alter the representation space, potentially affecting concept separability?

2. Approximation in Magnitude and Orthogonality of Representations: In cases where the representations for binary and hierarchical concepts are not exactly separable, do you have methods to measure or improve the alignment with the proposed theoretical structure (e.g., adjusting vector magnitudes or enforcing orthogonality)?

3. Impact of Hierarchical Orthogonality on Model Behavior: Could you provide more examples or case studies on how hierarchical orthogonality impacts the model’s downstream tasks? For instance, does this structure improve classification accuracy or clustering quality in hierarchical classification tasks?

---

> ### Author Response · Authors · 2024-11-25
> **Response**
>
> Thank you for your thoughtful review!
>
> With respect to your questions:
> 1. Park et al. (2024) provide a scheme for estimating the causal inner product using the inverse covariance matrix of the word unembeddings, and give evidence that this works well (in the sense of respecting semantics) for multiple language models. We simply follow their construction. Empirically, we find this again works well for multiple language models (i.e., we see the theoretically predicted constant magnitudes and hierachical orthogonalities, and this holds for the causal inner product but not the simple Euclidean one; see Figure 12).
> 2. To clarify: the geometric structure is a thing we derive as a consequence of the softmax distribution. It is a consequence, not a desiderata! Accordingly, we do not attempt to enforce it. Strictly speaking, the theory doesn’t make predictions about what happens when exact separability fails. However, we observe in practice that the degree of cossine similarity seems reflective of intutive human judgements about semantic closeness.
> 3. To clarify: the `downstream task’ here is language modeling :) The geometric representation of hierarchy emerges as a necessary component of modeling the structure of language using the softmax distribution. We anticipate that the hierachical structure uncovered here will be useful, in particular, for interpretability and control of LLMs---see discussion starting at line 525. However, building up such tools is very involved, and well beyond the scope of the present paper, which focuses on deriving and validating the structure.

---

### Official Review · Reviewer_GuHR · 2024-11-04

**Soundness:** 3
**Presentation:** 3
**Contribution:** 3
**Rating:** 6
**Confidence:** 3

**Summary:**

The paper generalises the idea that binary concepts can be represented as directions of their vectors in a vector space. The authors argue that this idea fails when concepts are not binary, e.g. we have mammals versus birds and each of these has many elements therein. They they construct a new theory that for such manifold concepts one needs a polytope in a vector space. They devise a nice theory,  generalising the notion of linear combinations over binary concepts to these polytopes and experiment with these findings using the vector spaces formed by Gemma and LLaMA.

**Strengths:**

1- Introducing a nice mathematically solid theory into the vector representations learnt by large language models.
2- Finding a novel way to represent ontological hierarchies of concepts in the spaces learnt by LLMs.
3- Solid maths and theory

**Weaknesses:**

Certain concepts are not well defined:

1- Please provide a clear definition of the unembedding space early in the paper, explaining its role in their theoretical framework.
2- Please provide concrete examples of concepts that are and are not causally separable, and explain how this distinction is crucial for their theoretical framework. The notion of causally separable introduced in the paper at the moment is vague, many concepts can be imagined to be causally separable. On the other hand, it is not clear at all what is not causally separable? I think the authors want to use this dichotomy to categorize entities under different general labels, but the definition is not helping.
3-  Do please provide a brief explanation of the causal inner product within their paper, highlighting its relevance to their work, rather than relying solely on the external reference. At the moment, the notion of a causal inner product is not clear. The authors refer to the work of Park et al, but I am not sure why and I am not sure what is taken from it.
4- The dataset on which the theory is testes is rather small (less than 600 nouns and less than 400 verbs). It does provide a proof of concept, but cannot be used as a large scale validation.  I unfortunately do not know what is mainstream in the field. I suggest the authors find a dataset used in knowledge graph literature and test on it.
5- I wonder why the theory has only been tested on nouns and verbs. In other words, it is not clear what exactly are concepts. I would like to suggest the authors explicitly define what they consider to be a "concept" in the context of their work, and discuss whether their theory could be applied to other parts of speech or more abstract concepts.
6- Please discuss how your approach compares to or could be integrated with existing work on knowledge graphs and ontologies, and how this might enhance the applicability of their theory. the paper needs a better theory of concepts to start from and to base itself on. I feel there is a good argument to be made by relating it to knowledge graphs and ontologies rather than working with a subset of WordNet.

**Questions:**

can you please answer to the points in weaknesses? I would appreciate it.

---

> ### Author Response · Authors · 2024-11-25
> **Response**
>
> Thank you for your constructive review!
>
> 1. With respect to the experimental evaluation, we want to emphasize that we are testing on (more than) 500 noun **concepts** and 300 verb **concepts**. These correspond to synsets in wordnet. The collection for each one actually contains at least 50 tokens. Across all of the concepts, we use every word in the wordnet hierarchy (that is also in the LLM vocabulary). WordNet provides a hierarchy for nouns and verbs; the reason that we tested on only these is that it is the entirety of the data! We also emphasize that the ordinary convention for establishing and testing linear representations focuses on just a single concept at a time! The experiments in this paper are, as far as we know, the largest scale effort to test the existence of linear representations of known concepts (so, excluding unsupervised discovery methods such as sparse autoencoders).
>
>
> 2. With respect to “what are concepts”: we formalize this (very generally) as latent variables that could in principle be manipulated (lines 119-125). Much of the paper (indeed, the study of representations generally) is about specializing this definition to something that allows deriving precise mathematical properties and empirical study. To that end, we specialize to binary concepts describing whether the output has a given attribute (i.e., is_animal, not_animal) and, ultimately, study the representations of such concepts using sets of words that exhibit the attribute (see beginning of section 3).
>
>
> 3. A connection to knowledge graphs and ontologies sounds like a very interesting connection! What papers did you have in mind? Note though that the main contribution of this paper is understanding the representation structure of hierarchy and categorical concepts in LLMs. Our aim is scientific. That is: we start with a hypothesis (the existence of linear representations), mathematize it, and use this to derive consequences that are in principle falsifiable (Theorem 4 and 8). The purpose of the experiments is to check empirically if these consequences hold. Accordingly, working with a relatively simple testbed---hierarchies such as dog is a mammal is an animal---is desirable because it allows for precise testing of the relevant hypothesis.
>
>
> 4. With respect to expanding the exposition on the things we’re using from previous work: we agree that, as much as possible, papers should be self-contained. However, this is a little bit tricky because we’re building on a paper that mainly aimed at getting definitions `right’---this involves a lot of mathematical detail that can be somewhat distracting for our purposes (e.g., the distinction between viewing the unembedding space as a vector space or an affine space). As such, we have reiterated their conclusions without trying to reproduce their motivations. However, we will add futher examples where the concepts are introduced as you suggest, and additional details to the appendix.
>
> In particular,
>
> - For the unembedding space, as mentioned in lines 107-111, the unembedding vectors are the token embeddings in the softmax layer that are multiplied to get the logits before applying the softmax function. During the training, the models focus on the softmax distribution, not any geometric relationship between the unembedding vectors.
>
> - With respect to causally inseparable concepts, consider first present_tense=>future_tense and present_tense=>past_tense. These concepts are not freely manipulable. As a second example, consider English=>German and LowerCase=>UpperCase---German and English have different capitalization rules, so these cannot be varied freely.
>
> - Concerning causal inner product, the geometric notions, such as the L2 norm in Definition 5 and orthogonality in Theorem 8, depend on the inner product in the embedding space. Then, we need a `suitable’ inner product for studying semantics. Park et al. argue that the causal inner product has a number of desirable properties for this purpose (and, in particular, the Euclidean inner product does not) so we build on their construction. Notice that in Figure 12, we explicitly show how the WordNet noun hierarchy is not encoded orthogonally using the naive Euclidean inner product.

---

> > ### Comment · Reviewer_GuHR · 2024-11-26
> > **response to the author response (of the original review)**
> >
> > Hi there, I would like to acknowledge that I have read the authors' response. Many thanks for providing  a thorough response. I agree with the authors that previous  theoretical work should be usable in later papers where empirical work is the main contribution. It would have been great if the definitions where, nonetheless, added in a self-containing manner. For instance, in the response above you do provide a better version of them.
> >
> > With regards to the WordNet dataset,  I still feel given the size of the vocabulary of English, this is still small. Also,  I feel it is the number of words that should matter rather than the number of tokens.
> >
> > Since the focus of this paper is empirical, a large scale validation would have been good to have at hand. Sweeping the simple English wiki will provide a much larger set of concepts to work with. Also, a tool such as  ConceptNet could also be used. I hope this helps.

---

> > > ### Author Response · Authors · 2024-11-26
> > > **Response to the reviewer's response**
> > >
> > > Thank you for your response!
> > >
> > > First, we want to emphasize that the present paper is not primarily empirical. The contribution is about formalizing a vector notion of linear representation, deriving precise consequences of this formalization, and then assessing empirically whether these consequences do in fact hold. In particular, the aim of the experiments here is *scientific*; that is, we are testing the theoretical predictions. As such, the aim is to measure specific phenomena in a tightly controlled setting that could be expected to fail if indeed the linear representation hypothesis is false. That is, the nature of the evidence here is *not* "we observe an unexplained but widespread phenomenon", it's "we make a precise prediction (based on a mathematical formalization) and find that it holds". From this perspective, the advantage of WordNet is that it provides a large curated collection of common hierarchical relationships (which we might reasonably expect an LLM to have learned). Spot-checking ConceptNet (which we were not previously familiar with---thank you for the pointer) doesn't seem helpful in this regard; e.g., the hyponym structure for "dog" is either included in WordNet (mammal, pet) or seems deeply idiosyncratic ("man's best friend"). Similarly, it's not clear to us what you have in mind by "sweeping" Wikipedia.
> > >
> > > It is also worth emphasizing that the (mathematical) results in the paper rely critically on the duality between sentence and word representations in LLMs.

---

### Official Review · Reviewer_yqwm · 2024-11-05

**Soundness:** 4
**Presentation:** 4
**Contribution:** 4
**Rating:** 8
**Confidence:** 3

**Summary:**

This paper presents a theoretical and empirical analysis of how concept hierarchies are encoded in the geometry of a pretrained LM's unembedding layer. The theoretical part develops a rigorous formalization of several notions related to the linear representation hypothesis, such as linear representations, vector representations of binary concepts, polytope representations of categorical concepts, and the notion of hierarchy being represented via orthogonality. The empirical part uses this formalism to identify how the conceptual structure of the WordNet hierarchy is reflected in the linear representational geometry of two LMs (Gemma-2B and Llama-3-8B). The empirical results, along with clear visualizations, constitute convincing evidence that the developed theoretical formalism closely tracks the actual representation of a conceptual hierarchy in the (un)embedding layer.

**Strengths:**

**Originality:** The sequence of definitions (linear representations, vector representations of binary concepts, polytope representations of categorical concepts) culminating in proposed structure of hierarchical orthogonality (Theorem 8) is a novel perspective while at the same time very intuitively appealing. Among the related work I'm familiar with, hierarchical orthogonality is reminiscent of partial orthogonality (Jiang et al., 2023; https://arxiv.org/abs/2310.17611, cited in the submission), but the two notions are distinct enough

**Quality:** The paper is of exemplary quality. The empirical analysis validates all theoretical predictions.

**Clarity:** The paper is exceptionally clear, well-written, and easy to read.

**Significance:** The paper significantly advances our understanding of the unembedding layer, revealing a highly intuitive linear structure in the representational geometry of a conceptual hierarchy. While the analysis in its current form is limited to concepts/features that are expressed as single tokens in the vocabulary of the LM's tokenizer, the formalism of hierarchical orthogonality can potentially provide a theoretical foundation for the analysis of more complex concepts in early-middle layers, which has recently become the focus of work on detokenization/retokenization (https://transformer-circuits.pub/2022/solu/index.html., https://arxiv.org/abs/2305.01610), stages of inference (https://arxiv.org/abs/2406.19384) and the LM's "inner vocabulary" (https://arxiv.org/abs/2410.05864).

**Weaknesses:**

I don't see any substantive weaknesses.

An obvious point that might be made here is the limitation (also acknowledged by the authors) that the presented formalism and empirical results are only applicable to the unembedding layer, but in my view the unembedding layer is sufficiently complex and interesting to serve as an object of study in its own right.

**Questions:**

n/a

---

> ### Author Response · Authors · 2024-11-25
>
> Thank you for your review! We’re very much appreciate your comments, and are glad to see the message came across clearly. We are also hopeful that the results here can be extended to the intermediate layers as well.

---

> > ### Comment · Reviewer_yBgV · 2024-11-27
> >
> > Thank you for providing a thorough and thoughtful response. I appreciate the time and effort you have dedicated to addressing the concerns and suggestions raised.

---

### Meta-Review · Area_Chair_4RMM · 2024-12-19

**Metareview:**

**Summary**

This paper presents a theoretical and empirical analysis of how concept hierarchies are encoded in the geometry of a pretrained LM's unembedding layer. The theoretical part develops a rigorous formalization of several notions related to the linear representation hypothesis, such as linear representations, vector representations of binary concepts, polytope representations of categorical concepts, and the notion of hierarchy being represented via orthogonality. The empirical part uses this formalism to identify how the conceptual structure of the WordNet hierarchy is reflected in the linear representational geometry of two LMs (Gemma-2B and Llama-3-8B). The empirical results, along with clear visualizations, constitute convincing evidence that the developed theoretical formalism closely tracks the actual representation of a conceptual hierarchy in the (un)embedding layer.



**Strengths**
- the paper presents a novel insight of undestanding how tranformers encode knowledge
- the paper is well written
- all empirical results validates theoretical predictions

**Weaknesses**

No big weaknesses

**Final remarks**

The paper clearly advaces knowledge in the field of LLMs. The three reviewers are fairly convinced.

**Additional Comments On Reviewer Discussion:**

The interaction has helped to clarify specific issues of the paper.

---

> ### Comment · Reviewer_yqwm · 2025-04-03
> **An area chair should know better**
>
> I just noticed that the summary section in this meta review is a verbatim copy of the summary written by me, Reviewer yqwm (https://openreview.net/forum?id=bVTM2QKYuA&noteId=2DrG25onBD), used without any attribution.
> I know that this is only a meta review for a pretty clear-cut decision and might appear trivial, but -- trivial or not, intentional or not -- passing off someone else's work as one's own is still plagiarism.

---

> > ### Comment · Area_Chair_4RMM · 2025-04-04
> > **Thanks for pointing it out.**
> >
> > You did such a great description of the content of the paper. It was already clear that it was yours.

---

### Decision · Program_Chairs · 2025-01-22

Accept (Oral)